# A Gradual Coarse-to-Fine Framework for Irregularly Sampled Multivariate Time Series Analysis

## Abstract

Irregularly sampled multivariate time series (ISMTS) are prevalent in reality. Most existing methods treat ISMTS as synchronized regularly sampled time series with missing values, neglecting that the irregularities are primarily attributed to variations in sampling rates. In this paper, we introduce a novel perspective that irregularity is essentially relative in some senses. With sampling rates artificially determined from low to high, an irregularly sampled time series can be transformed into a hierarchical set of relatively regular time series from coarse to fine. We observe that additional coarse-grained relatively regular series not only mitigate the irregularly sampled challenges but also incorporate broad-view temporal information, thereby serving as a valuable asset for representation learning. Therefore, following the philosophy of learning that Seeing the big picture first, then delving into the details, we present the **Mu**lti-**S**cale and Mult**i**-**C**orrelation Attention Network (MuSiCNet) combining multiple scales to iteratively refine the ISMTS representation. Specifically, within each scale, we explore time attention and frequency correlation matrices to aggregate intra- and inter-series information, naturally enhancing the representation quality with richer and more intrinsic details. While across adjacent scales, we employ a representation rectification method containing contrastive learning and reconstruction results adjustment to further improve representation consistency. MuSiCNet is an ISMTS analysis framework that competitive with SOTA in three mainstream tasks consistently, including classification, interpolation, and forecasting.

## 1 Introduction

Irregularly sampled multivariate time series (ISMTS) are ubiquitous in realistic scenarios, ranging from scientific explorations to societal interactions (Che et al., 2018; Shukla & Marlin, 2021; Sun et al., 2021; Agarwal et al., 2023; Yalavarthi et al., 2024). The causes of irregularities in time series collection are diverse, including sensor malfunctions, transmission distortions, cost-reduction strategies, and various external forces or interventions, etc. Such ISMTS data exhibit distinctive features including intra-series irregularity, characterized by inconsistent intervals between consecutive data points, and inter-series irregularity, marked by a lack of synchronization across multiple variables. The above characteristics typically result in the lack of alignment and uneven count of observations (Shukla & Marlin, 2020), invalidating the assumption of coherent fixed-dimensional feature space for most traditional time series analysis models.

Recent studies have attempted to address these challenges by treating ISMTS as synchronized, regularly sampled Normal Multivariate Time Series (NMTS) data with missing values, focusing on imputation strategies (Che et al., 2018; Yoon et al., 2018; Camino et al., 2019; Tashiro et al., 2021; Zhang et al., 2021c; Chen et al., 2022; Fan, 2022; Du et al., 2023). However, direct imputation is difficult, especially when sampling is sparse. Inaccurate imputation results can distort underlying relationships and introduce significant noise, which can greatly reduce the accuracy of analysis tasks (Zhang et al., 2021b; Wu et al., 2021; Agarwal et al., 2023; Sun et al., 2024). Latest developments circumvent imputation and aim to address these challenges by embracing the inherent continuity of time, thus preserving the continuous temporal dynamics dependencies within the ISMTS data. Despite these innovations, most methods above are merely solutions for intra-series irregularities,

such as Recurrent Neural Networks (RNNs) (De Brouwer et al., 2019; Schirmer et al., 2022; Agarwal et al., 2023)- and Neural Ordinary Differential Equations (Neural ODEs)-based methods (Kidger et al., 2020; Rubanova et al., 2019; Jhin et al., 2022; Jin et al., 2022) and the unaligned challenges presented by inter-series irregularities in multivariate time series remain unsolved.

Delving into the nature of irregularly sampled time series, we discover that the intra- and inter-series irregularities in ISMTS primarily arise from inconsistency in sampling rates within and across variables. We argue that irregularities are essentially relative in some senses and by artificially determined sampling rates from low to high, ISMTS can be transformed into a hierarchical set of relatively regular time series from coarse to fine. Taking a broader perspective, setting a lower and consistent sampling rate within an instance can synchronize sampling times across series and establish uniform time intervals within series. This approach can mitigate both types of irregularity and emphasize long-term dependencies. As shown in Fig.1, the coarse-grained scales 1 and 2 exhibit balanced placements for all variables in the instance and provide clearer overall trends. However, lower sampling rates may lead to information loss and sacrifice detailed temporal variations. Conversely, with a higher sampling rate as in scale $L$, more real observations contain rich information and prevent artificially introduced dependencies beyond original relations during training. Nonetheless, the significant irregularity in fine-grained scales poses a greater challenge for representation learning.

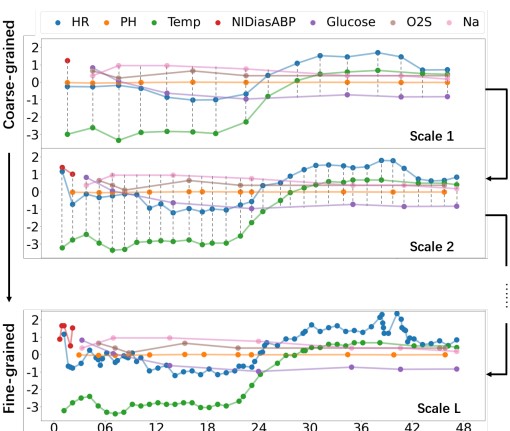

Figure 1: Comparative visualization of multi-scale time series data with various sampling rates. Scale $L$ depicts the original selected representative time series in the P12 Dataset to show the inter- and intra-series irregularities. Scale 1 to Scale $L-1$ illustrates the effect of applying different sampling rates from low to high.

To bridge this gap, we propose MuSiCNet—a **Mu**lti-**S**cale and Multi-**C**orrelation Attention Network—to iteratively optimize ISMTS representations from coarse to fine. Our approach begins by establishing a hierarchical set of coarse- to fine-grained series with sampling rates from low to high. **At each scale**, we employ a custom-designed encoder-decoder framework called multi-correlation attention network (CorrNet), for representation learning. Different from most existing methods that focus mainly on intra-series relationships, our CorrNet encoder (CorrE) captures embeddings of continuous time values by using an attention mechanism containing correlation matrices to aggregate both intra- and inter-series information. This approach is crucial not only because every observation in ISMTS, given the sparse sampling, is valuable for representation learning, but also due to the fact that correlated variables provide deeper insights for a given query. Therefore, we designed frequency correlation matrices using Lomb–Scargle Periodogram-based Dynamic Time Warping (LSP-DTW). This approach addresses the challenges of calculating correlations in ISMTS and re-weights the inter-series attention scores to better capture cross-series information. **Across scales**, we employ a representation rectification operation from coarse to fine to iteratively refine the learned representations with contrastive learning and reconstruction results adjustment methods. This ensures accurate and consistent representation and minimizes error propagation throughout the model.

Benefiting from the aforementioned designs, MuSiCNet explicitly learns multi-scale information, enabling good performance on widely used ISMTS datasets, thereby demonstrating its ability to capture relevant features for ISMTS analysis. Our main contributions can be summarized as follows:

- We find that irregularities in ISMTS are essentially relative in some senses and multi-scale learning helps balance coarse- and fine-grained information in ISMTS representation learning.

- We introduce CorrNet, an encoder-decoder framework designed to learn fixed-length representations for ISMTS. Notably, our proposed LSP-DTW can mitigate spurious correlations induced by irregularities in the frequency domain and effectively re-weight attention across sequences.

- We are not limited to a specific analysis task and attempt to propose a task-general model for ISMTS analysis, including classification, interpolation, and forecasting.

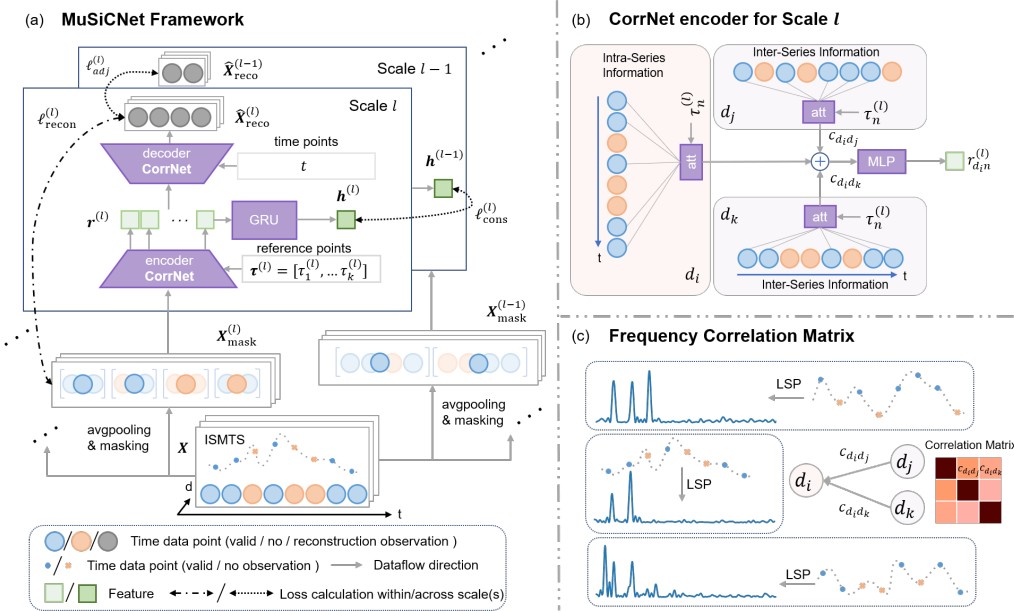

Figure 2: Overview of MuSiCNet framework, shown in (a), containing three main components for better representation learning, including hierarchical structure $\{X_{\text{mask}}^{(l)}\}_{l=1}^{L}$, representation learning using CorrNet within scale $\ell_{\text{cons}}^{(l)}$, and rectification operation across adjacent scales $\ell_{\text{recon}}^{(l)}$. (b) visualizes the encoding process in CorrNet for Scale $l$, which relies on $\tau_n^{(l)}$ to aggregates intra-series information, and then relies on $c_{d_i,(\cdot)}$ to fuse inter-series information from other variables for $d_i$-th dimension. (c) visualizes the calculation process of the correlation matrix, which transfers the time domain into the frequency domain with LSP, and then utilizes DTW to calculate the similarity weight.

## 2 RELATED WORK

**Irregularly Sampled Multivariate Time Series Analysis.** An effective approach for analyzing ISMTS hinges on the understanding of their unique properties. Most existing methods treat ISMTS as NMTS with missing values, such as Che et al. (2018); Yoon et al. (2018); Camino et al. (2019); Tashiro et al. (2021); Chen et al. (2022); Fan (2022); Du et al. (2023); Wang et al. (2024). However, most imputation-based methods may distort the underlying relationships, introducing unsuitable inductive biases and substantial noise due to incorrect imputation (Zhang et al., 2021b; Wu et al., 2021; Agarwal et al., 2023), ultimately compromising the accuracy of downstream tasks. Some other methods treat ISMTS as time series with discrete timestamps, aggregating all sample points of a single variable to extract a unified feature for each variable (Zhang et al., 2021b; Horn et al., 2020; Li et al., 2023). These methods can directly accept raw ISMTS data as input but often struggle to handle the underlying relationships within the time series. Recent progress seeks to overcome these challenges by recognizing and utilizing the inherent continuity of time, thereby maintaining the ongoing temporal dynamics present in ISMTS data (De Brouwer et al., 2019; Rubanova et al., 2019; Kidger et al., 2020; Schirmer et al., 2022; Jhin et al., 2022; Chowdhury et al., 2023).

Despite these advancements, existing methods mainly suffer from two main drawbacks, they primarily address intra-series irregularity while overlooking the alignment issues stemming from inter-series irregularity, and 2) they rely on assumptions tailored to specific downstream tasks (Yalavarthi et al., 2024; Wang et al., 2024), hindering their ability to consistently perform well across various ISMTS tasks.

**Multi-scale Modeling.** Multi-scale and hierarchical approaches have demonstrated their utility across various fields, including computer vision (CV) (Fan et al., 2021; Zhang et al., 2021a), natural language processing (NLP) (Nawrot et al., 2021; Zhao et al., 2021), and time series analysis (Chen et al., 2021; Shabani et al., 2022; Cai et al., 2024). Most recent innovations in the time series analysis domain have seen the integration of multi-scale modules into the Transformer architecture to enhance analysis capabilities Shabani et al. (2022); Liu et al. (2021) and are designed for NMTS.

Nevertheless, the application of multi-scale modeling specifically designed for ISMTS data, and the exploitation of information across scales, remain less unexplored. As far as we know, Singh et al. (2019) and Zhang et al. (2023) are among the earlier works on multi-level ISMTS learning. Singh et al. (2019) addresses multi-resolution signal issues by distributing signals across specialized branches with different resolutions, where each branch employs a Flexible Irregular Time Series Network (FIT) to process high- and low-frequency data separately. Zhang et al. (2023), on the other hand, is a transformer-based model that stacks multiple Warpformer layers to produce multi-scale representations, combining them via residual connections to support downstream tasks. These works typically focus on either specific tasks or particular model architectures. In contrast, our design philosophy originates from ISMTS characteristics rather than being tied to a specific feature extraction network structure. Warpformer emphasizes designing a specific network architecture but involves high computational costs and requires manually balancing the trade-off between the number of scales and the dataset. These are challenges that our MuSiCNet avoids entirely.

## 3 PROPOSED MUSICNET FRAMEWORK

As previously mentioned, our work aims to learn ISMTS representation for further analysis tasks by introducing MuSiCNet, a novel framework designed to balance coarse- and fine-grained information across different scales. The overall model architecture illustrated in Fig.2(a) indicates the effectiveness of MuSiCNet can be guaranteed to a great extent by 1) **Hierarchical Structure**. 2) **Representation Learning Using CorrNet Within Scale**. 3) **Rectification Across Adjacent Scales**. We will first introduce problem formulation and notations of MuSiCNet and then discuss key points in the following subsections.

### 3.1 PROBLEM FORMULATION

Our goal is to learn a nonlinear embedding function $f_\theta$, such that the set of ISMTS data $\mathcal{X} = \{\boldsymbol{X}_1, \cdots, \boldsymbol{X}_N\}$ can map to the best-described representations for further ISMTS analysis including both supervised and unsupervised tasks. We denote $\boldsymbol{X}_n \in \mathbb{R}^{T_n \times D}$ as a D-dimensional instance with the length of observation $T_n$. Specifically, the $d$-th dimension in instance $n$ can be treated as a tuple $\boldsymbol{X}_{dn} = (\boldsymbol{x}_{dn}, \boldsymbol{t}_{dn})$ where the length of observations is $T_{dn}$. $\boldsymbol{x}_{dn} = [x_{1dn}, \cdots x_{T_{dn}dn}]$ is the list of observations and the list of corresponding observed timestamps is $\boldsymbol{t}_{dn} = [t_{1dn}, \cdots t_{T_{dn}dn}]$. *We drop the data case index $n$ for brevity when the context is clear.*

### 3.2 CORRNET ARCHITECTURE WITHIN SCALE

**Multi-Correlation Attention Module.** In this subsection, we elaborate on the Multi-Correlation Attention module. Time attention has proven effective for ISMTS learning (Shukla & Marlin, 2021; Horn et al., 2020; Chowdhury et al., 2023; Yu et al., 2024). Many existing methods mainly focus on capturing interactions between observation values and their corresponding sampling times within a single variable. However, due to the potential sparse sampling in ISMTS, observations from all variables are valuable and need to be considered.

To address this, we use irregularly sampled time points and corresponding observations from all variables within a sample as keys and values to produce fixed-dimensional representations at the query time points. The importance of each variable cannot be uniform for a given query and variables that provide more valuable information should receive more attention. Therefore, we designed frequency correlation matrices to re-weight the inter-series attention scores, enhancing the representation learning process.

In general, as illustrated in Fig.2(b), taking ISMTS $\boldsymbol{X}$ as input, the CorrNet Encoder $\mathrm{CorrE}(\cdot)$ generates multi-time attention embedding as follows:

$$\begin{aligned} \mathrm{CorrE}(\boldsymbol{Q}_T, \boldsymbol{K}_T, \boldsymbol{X}) &= \boldsymbol{A}_T \boldsymbol{X} \boldsymbol{C}_T \\ \boldsymbol{A}_T &= \mathrm{softmax}(\boldsymbol{Q}_T \boldsymbol{K}_T / d_r) \end{aligned} \tag{1}$$

where the calculation of $\boldsymbol{A}_T \in \mathbb{R}^{K \times T}$ is based on a time attention mechanism with query $\boldsymbol{Q}_T \in \mathbb{R}^{K \times K}$ and key $\boldsymbol{K}_T \in \mathbb{R}^{K \times T}$ (Vaswani et al., 2017). Since more attention should be paid to correlated variables for a given query which can provide more valuable knowledge. Therefore,

different input dimensions should utilize various weights of time embeddings through the correlation matrix $\boldsymbol{C}_T \in \mathbb{R}^{D \times D}$, and we will introduce it in the following Correlation Extraction paragraph.

Since the continuous function defined by the CorrE module is incompatible with neural network architectures designed for fixed-dimensional vectors or discrete sequences, following the method in Shukla & Marlin (2021), we generate an output representation by materializing its output at a pre-defined set of reference time points $\boldsymbol{\tau} = [\tau_1, \cdots, \tau_K]$. This process transforms the continuous output into a fixed-dimensional vector or a discrete sequence, thereby making it suitable for subsequent neural network processing.

**Correlation Extraction.** The correlation matrix is essential for deriving reliable and consistent correlations within ISMTS, which must be robust to the inherent challenges of variable sampling rates and inconsistent observation counts at each timestamp in ISMTS. Most existing distance measures, such as Euclidean distance, Dynamic Time Warping (DTW) (Berndt & Clifford, 1994), and Optimal Transport / Wasserstein Distance (Villani et al., 2009), risk generating spurious correlations in the context of irregularly sampled time series. This is due to their dependence on the presence of both data points for the similarity measurement, and the potential for imputation to introduce unreliable information before calculating similarity which will be explored further in Section 4.4 of our experiments.

At an impasse, the Lomb-Scargle Periodogram (LSP) (Lomb, 1976; Scargle, 1982) provides en-lightenment to address this issue. LSP is a well-known algorithm for generating a power spectrum and detecting the periodic component in irregularly sampled time series. It extends the *Fourier periodogram* approach to accommodate irregularly sampled scenarios (VanderPlas, 2018) eliminat-ing the need for interpolation or imputation. This makes LSP a great tool for simplifying ISMTS analysis. Compared to existing methods, measuring the similarity between discrete raw observations, LSP-DTW, an implicit continuous method, utilizes inherent periodic characteristics and provides global information to measure the similarity.

As demonstrated in Fig.2(c), we first convert ISMTS into the frequency domain using LSP and then apply DTW to evaluate the distance between variables. The correlation between $\boldsymbol{X}_{d_i}$ and $\boldsymbol{X}_{d_j}$ is:

$$c_{d_i d_j} = \text{DTW}\left(\text{LSP}(\boldsymbol{X}_{d_i}), \text{LSP}(\boldsymbol{X}_{d_j})\right) = \min_{\pi} \sum\nolimits_{(m,n) \in \pi} \left(\text{LSP}(\boldsymbol{X}_{d_i})[m] - \text{LSP}(\boldsymbol{X}_{d_j})[n]\right)^2 \quad (2)$$

where $\pi$ is the search path of DTW. We calculate the correlation matrix $C_T$ by iteratively performing the aforementioned step for an instance.

Notably, we compute the correlation matrix using LSP-DTW only once per instance, without iteratively applying it, and it is not calculated in model training or inference.

**Encoder-Decoder Framework.** Drawing inspiration from notable advances in NLP and CV, our core network, CorrNet employs time series masked modeling, which learns effective time series representations to facilitate various downstream analysis tasks. It is a framework consisting of an encoder-decoder architecture based on continuous-time interpolation. At each scale $l$, CorrE learns a set of latent representations $\boldsymbol{r}^{(l)} = [r_1, \cdots, r_K]$ defined at $K$ reference time points on the randomly masked ISMTS. We further employ CorrNet Decoder (CorrD), a simplified CorrE (without correlation matrix), to produce the reconstructed output $\hat{\boldsymbol{X}}_{\text{reco}}^{(l)}$, using the input time point sequence $\boldsymbol{t}^{(l)}$ as reference points. We iteratively apply the same CorrNet at each scale. Here, we emphasize that all scales share a single encoder that can reduce the model complexity and keep feature extraction consistency for various scales.

We measure the reconstruction accuracy using the Mean Squared Error (MSE) between the recon-structed values and the original ones at each timestamp and calculate the MSE loss specifically for the masked timestamps, as expressed in the following equation

$$\ell_{\text{recon}}^{(l)} = \sum\nolimits_{n=1}^{N} \left\| \boldsymbol{M}^{(l)} \odot \left( \left( \hat{\boldsymbol{X}}_{\text{reco}}^{(l)} \right)_n - \boldsymbol{X}_n^{(l)} \right) \right\|_2^2 \quad (3)$$

where $\boldsymbol{M}^{(l)}$ is the mask for the $l$-th scale, $\odot$ is the Hadamard product.

### 3.3 RECTIFICATION STRATEGY ACROSS SCALES

Following the principle that adjacent scales exhibit similar representations and coarse-grained scales contain more long-term information, the rectification strategy is a key component of our MuSiCNet framework. We implement a dual rectification strategy across adjacent scales to enhance representation learning. We start by generating a hierarchical set of relatively regular time series from coarse to fine by

$$\boldsymbol{X}_{\text{multi}} = \boldsymbol{M}^{(l)} \odot \left(\text{AvgPooling}_L\left(\boldsymbol{X}\right)\right) = \{\boldsymbol{X}_{\text{mask}}^{(1)}, \cdots, \boldsymbol{X}_{\text{mask}}^{(L)}\} \tag{4}$$

While the coarse-grained series ignores detailed variations for high-frequency signals and focuses on much clearer broad-view temporal information, the fine-grained series retains detailed variations for frequently sampled series. As a result, iteratively using coarse-grained information for fine-grained series as a strong structural prior can benefit ISMTS learning.

Firstly, the reconstruction results at scale $l$ is designed to align closely with the results at the $(l-1)$-th scale, that is to say, the reconstruction results at scale $(l-1)$ can be used to adjust the results at scale $l$ using MSE,

$$\ell_{\text{adj}}^{(l)} = \sum_{n=1}^{N} \left\| \left(\text{AvgPooling}_l(\hat{\boldsymbol{X}}_{\text{reco}}^{(l)})\right)_n - \left(\hat{\boldsymbol{X}}_{\text{reco}}^{(l-1)}\right)_n \right\|_2^2 \tag{5}$$

Secondly, contrastive learning is leveraged to ensure coherence between adjacent scales. Pulling these two representations between adjacent scales together and pushing other representations within the batch $\mathcal{B}$ apart, not only facilitates the learning of within-scale representations but also enhances the consistency of cross-scale representations. Taking into consideration that the dimensions of $\boldsymbol{r}^{(l)}$ and $\boldsymbol{r}^{(l-1)}$ are different, we employ a GRU Network as a decoder to uniform dimension as $\boldsymbol{h}^{(l)}$ and $\boldsymbol{h}^{(l-1)}$ before contrastive learning.

$$\ell_{\text{cons}}^{(l)} = -\sum_{i=1}^{N} \log \frac{\exp\left(\boldsymbol{h}_i^{(l)} \cdot \boldsymbol{h}_i^{(l-1)}\right)}{\sum_{j=1}^{\mathcal{B}} \left(\exp\left(\boldsymbol{h}_i^{(l)} \cdot \boldsymbol{h}_j^{(l-1)}\right) + \mathbb{I}_{[i \neq j]} \exp\left(\boldsymbol{h}_i^{(l)} \cdot \boldsymbol{h}_j^{(l)}\right)\right)} \tag{6}$$

where the $\mathbb{I}$ is the indicator function.

The advantage of the two operations lies in their ability to ensure a consistent and accurate representation of the data at different scales. This strategy significantly improves the model's ability to learn representations from ISMTS data, which is essential for tasks requiring detailed and accurate time series analysis. Last but not least, this method ensures that the model remains robust and effective even when dealing with data at varying scales, making it versatile for diverse applications.

## 4 EXPERIMENT

In this section, we demonstrate the effectiveness of the MuSiCNet framework for time series classification, interpolation and forecasting. *Notably, for each dataset, the window size is initially set to $1/4$ of the time series length and then halved iteratively until the majority of the windows contain at least one observation.* Our results are based on the mean and standard deviation values computed over 5 independent runs. **Bold** indicates the best performer, while underline represents the second best. Due to the page limitation, we provide more detailed setup for experiments in the Appendix.

### 4.1 TIME SERIES CLASSIFICATION

**Datasets and experimental settings.** We use real-world datasets including healthcare and human activity for classification. (1) **P19** (Reyna et al., 2020) with missing ratio up to $94.9\%$, includes $38,803$ patients that are monitored by $34$ sensors. (2) **P12** (Goldberger et al., 2000) records temporal measurements of $36$ sensors of $11,988$ patients in the first 48-hour stay in ICU, with a missing ratio of $88.4\%$. (3) **PAM** (Reiss & Stricker, 2012) contains $5,333$ segments from $8$ activities of daily living that are measured by $17$ sensors and the missing ratio is $60.0\%$. *Importantly, P19 and P12 are **imbalanced** binary label datasets.*

Here, we follow the common setup by randomly splitting the dataset into training ($80\%$), validation ($10\%$), and test ($10\%$) sets and the indices of these splits are fixed across all methods. Consistent with prior researches, we evaluate the performance of our framework on classification tasks using the area

Table 1: Comparison with the baseline methods on ISMTS **classification** task.

| Methods | P19 | | P12 | | PAM | | | |
|---|---|---|---|---|---|---|---|---|
| | AUROC | AUPRC | AUROC | AUPRC | Accuracy | Precision | Recall | F1 score |
| Transformer | $80.7 \pm 3.8$ | $42.7 \pm 7.7$ | $83.3 \pm 0.7$ | $47.9 \pm 3.6$ | $83.5 \pm 1.5$ | $84.8 \pm 1.5$ | $86.0 \pm 1.2$ | $85.0 \pm 1.3$ |
| Trans-mean | $83.7 \pm 1.8$ | $45.8 \pm 3.2$ | $82.6 \pm 2.0$ | $46.3 \pm 4.0$ | $83.7 \pm 2.3$ | $84.9 \pm 2.6$ | $86.4 \pm 2.1$ | $85.1 \pm 2.4$ |
| GRU-D | $83.9 \pm 1.7$ | $46.9 \pm 2.1$ | $81.9 \pm 2.1$ | $46.1 \pm 4.7$ | $83.3 \pm 1.6$ | $84.6 \pm 1.2$ | $85.2 \pm 1.6$ | $84.8 \pm 1.2$ |
| SeFT | $81.2 \pm 2.3$ | $41.9 \pm 3.1$ | $73.9 \pm 2.5$ | $31.1 \pm 4.1$ | $67.1 \pm 2.2$ | $70.0 \pm 2.4$ | $68.2 \pm 1.5$ | $68.5 \pm 1.8$ |
| mTAND | $84.4 \pm 1.3$ | $50.6 \pm 2.0$ | $84.2 \pm 0.8$ | $48.2 \pm 3.4$ | $74.6 \pm 4.3$ | $74.3 \pm 4.0$ | $79.5 \pm 2.8$ | $76.8 \pm 3.4$ |
| IP-Net | $84.6 \pm 1.3$ | $38.1 \pm 3.7$ | $82.6 \pm 1.4$ | $47.6 \pm 3.1$ | $74.3 \pm 3.8$ | $75.6 \pm 2.1$ | $77.9 \pm 2.2$ | $76.6 \pm 2.8$ |
| DGM$^2$-O | $86.7 \pm 3.4$ | $44.7 \pm 11.7$ | $84.4 \pm 1.6$ | $47.3 \pm 3.6$ | $82.4 \pm 2.3$ | $85.2 \pm 1.2$ | $83.9 \pm 2.3$ | $84.3 \pm 1.8$ |
| MTGNN | $81.9 \pm 6.2$ | $39.9 \pm 8.9$ | $74.4 \pm 6.7$ | $35.5 \pm 6.0$ | $83.4 \pm 1.9$ | $85.2 \pm 1.7$ | $86.1 \pm 1.9$ | $85.9 \pm 2.4$ |
| Raindrop | $87.0 \pm 2.3$ | $51.8 \pm 5.5$ | $82.8 \pm 1.7$ | $44.0 \pm 3.0$ | $88.5 \pm 1.5$ | $89.9 \pm 1.5$ | $89.9 \pm 0.6$ | $89.8 \pm 1.0$ |
| Warpformer | $\underline{88.8} \pm 1.7$ | $\underline{55.2} \pm 3.9$ | $83.4 \pm 0.9$ | $47.2 \pm 3.7$ | $94.3 \pm 0.6$ | $95.8 \pm 0.8$ | $94.8 \pm 1.0$ | $95.2 \pm 0.6$ |
| ViTST | $\mathbf{89.2} \pm 2.0$ | $\mathbf{53.1} \pm 3.4$ | $\underline{85.1} \pm 0.8$ | $\underline{51.1} \pm 4.1$ | $\underline{95.8} \pm 1.3$ | $\underline{96.2} \pm 1.3$ | $\underline{96.1} \pm 1.1$ | $\underline{96.5} \pm 1.2$ |
| **MuSiCNet** | $86.8 \pm 1.4$ | $45.4 \pm 2.7$ | $\mathbf{86.1} \pm 0.4$ | $\mathbf{54.1} \pm 2.2$ | $\mathbf{96.3} \pm 0.7$ | $\mathbf{96.9} \pm 0.6$ | $\mathbf{96.9} \pm 0.5$ | $\mathbf{96.8} \pm 0.5$ |

Table 2: Comparison with the baseline methods on ISMTS **interpolation** task on PhysioNet.

| Model | Mean Squared Error ($\times 10^{-3}$) | | | | |
|---|---|---|---|---|---|
| Observed % | 50% | 60% | 70% | 80% | 90% |
| RNN-VAE | $13.418 \pm 0.008$ | $12.594 \pm 0.004$ | $11.887 \pm 0.005$ | $11.133 \pm 0.007$ | $11.470 \pm 0.006$ |
| L-ODE-RNN | $8.132 \pm 0.020$ | $8.140 \pm 0.018$ | $8.171 \pm 0.030$ | $8.143 \pm 0.025$ | $8.402 \pm 0.022$ |
| L-ODE-ODE | $6.721 \pm 0.109$ | $6.816 \pm 0.045$ | $6.798 \pm 0.143$ | $6.850 \pm 0.066$ | $7.142 \pm 0.066$ |
| mTAND-Full | $\underline{4.139} \pm 0.029$ | $\underline{4.018} \pm 0.048$ | $\underline{4.157} \pm 0.053$ | $\underline{4.410} \pm 0.149$ | $\underline{4.798} \pm 0.036$ |
| **MuSiCNet** | $\mathbf{0.918} \pm 0.025$ | $\mathbf{0.919} \pm 0.064$ | $\mathbf{0.938} \pm 0.014$ | $\mathbf{0.992} \pm 0.008$ | $\mathbf{0.965} \pm 0.008$ |

under the receiver operating characteristic curve (AUROC) and the area under the precision-recall curve (AUPRC) for the P12 and P19 datasets, given their imbalanced nature. For the nearly balanced PAM dataset, we employ Accuracy, Precision, Recall, and F1 Score. For all of the above metrics, higher results indicate better performance.

**Main Results of classification.**    We compare MuSiCNet with ten state-of-the-art irregularly sampled time series classification methods, including Transformer (Vaswani et al., 2017), Trans-mean, GRU-D (Che et al., 2018), SeFT (Horn et al., 2020), and mTAND (Shukla & Marlin, 2021), IP-Net (Shukla & Marlin, 2018), DGM$^2$-O(Wu et al., 2021), MTGNN (Wu et al., 2020), Raindrop (Zhang et al., 2021b), ViTST (Li et al., 2023) and Warpformer (Zhang et al., 2023). Since mTAND is proven superior over various recurrent models, such as RNNImpute (Che et al., 2018), Phased-LSTM (Neil et al., 2016) and ODE-based models like LATENT-ODE and ODE-RNN (Chen et al., 2018), we focus our comparisons on mTAND and do not include results for the latter model.

As indicated in Table 1, MuSiCNet demonstrates good performance across three benchmark datasets, underscoring its effectiveness in typical time series classification tasks. Notably, in binary classification scenarios, MuSiCNet surpasses the best-performing baselines on the P12 dataset by an average of 1.0% in AUROC and 3.0% in AUPRC. For the P19 dataset, while our performance is competitive, MuSiCNet stands out due to its lower time and space complexity compared to ViTST. ViTST converts 1D time series into 2D images, potentially leading to significant space inefficiencies due to the introduction of extensive blank areas, especially problematic in ISMTS. In the more complex task of 8-class classification on the PAM dataset, MuSiCNet surpasses current methodologies, achieving a 0.5% improvement in accuracy and a 0.7% increase in precision.

Notably, the *consistently low standard deviation* in our results indicates that MuSiCNet is a reliable model. Its performance remains steady across varying data samples and initial conditions, suggesting a strong potential for generalizing well to new, unseen data. This stability and predictability in performance enhance the confidence in the model's predictions, which is particularly crucial in sensitive areas such as medical diagnosis in clinical settings.

Table 3: Experimental results for **forecasting** next three time steps. $-$ indicates no published results.

| Methods | USHCN | MIMIC-III | Physionet12 |
|---|---|---|---|
| DLinear+ | $0.347 \pm 0.065$ | $0.691 \pm 0.016$ | $0.380 \pm 0.001$ |
| NLinear+ | $0.452 \pm 0.101$ | $0.726 \pm 0.019$ | $0.382 \pm 0.001$ |
| Informer+ | $0.320 \pm 0.047$ | $0.512 \pm 0.064$ | $0.347 \pm 0.001$ |
| FedFormer+ | $2.990 \pm 0.476$ | $1.100 \pm 0.059$ | $0.455 \pm 0.004$ |
| NeuralODE-VAE | $0.960 \pm 0.110$ | $0.890 \pm 0.010$ | $-$ |
| GRU-Simple | $0.750 \pm 0.120$ | $0.820 \pm 0.050$ | $-$ |
| GRU-D | $0.530 \pm 0.060$ | $0.790 \pm 0.060$ | $-$ |
| T-LSTM | $0.590 \pm 0.110$ | $0.620 \pm 0.050$ | $-$ |
| mTAND | $0.300 \pm 0.038$ | $0.540 \pm 0.036$ | $0.315 \pm 0.002$ |
| GRU-ODE-Bayes | $0.430 \pm 0.070$ | $0.480 \pm 0.480$ | $0.329 \pm 0.004$ |
| Neural Flow | $0.414 \pm 0.102$ | $0.490 \pm 0.004$ | $0.326 \pm 0.004$ |
| CRU | $0.290 \pm 0.060$ | $0.592 \pm 0.049$ | $0.379 \pm 0.003$ |
| GraFITi | $\underline{0.272} \pm 0.047$ | $\mathbf{0.396} \pm 0.030$ | $\mathbf{0.286} \pm 0.001$ |
| MuSiCNet | $\mathbf{0.268} \pm 0.038$ | $\underline{0.475} \pm 0.031$ | $\underline{0.312} \pm 0.000$ |

## 4.2 TIME SERIES INTERPOLATION

**Datasets and experimental settings.** **PhysioNet** (Silva et al., 2012) consists of 37 variables extracted from the first 48 hours after admission to the ICU. We use all $8,000$ instances for interpolation experiments whose missing ratio is $78.0\%$.

We randomly split the dataset into a training set, encompassing $80\%$ of the instances, and a test set, comprising the remaining $20\%$ of instances. Additionally, $20\%$ of the training data is reserved for validation purposes. The performance evaluation is conducted using MSE, where lower values indicate better performance.

**Main Results of Interpolation.** For the interpolation task, we compare it with RNN-VAE, L-ODE-RNN (Chen et al., 2018), L-ODE-ODE (Rubanova et al., 2019), mTAND-full.

For the interpolation task, models are trained to predict or reconstruct values for the entire dataset based on a selected subset of available points. Experiments are conducted with varying observation levels, ranging from $50\%$ to $90\%$ of observed points. During test time, models utilize the observed points to infer values at all time points in each test instance.

As illustrated in Table 2, MuSiCNet demonstrates superior performance, highlighting its effectiveness in time series interpolation. This can be attributed to its ability to interpolate progressively from coarse to fine, aligning with the intuition of multi-resolution signal approximation (Mallat, 1989).

## 4.3 TIME SERIES FORECASTING

**Datasets and Experimental Settings.** (1) **USHCN** (Menne et al., 2015) is an artificially preprocessing dataset containing measurements of 5 variables from 1280 weather stations in the USA. The missing ratio is $78.0\%$. (2) **MIMIC-III** (Johnson et al., 2016) are dataset that rounded the recorded observations into 96 variables, 30-minute intervals and only use observations from the 48 hours after admission. The missing ratio is $94.2\%$. (3) **Physionet12** (Silva et al., 2012) comprises medical records from $12,000$ ICU patients. It includes measurements of 37 vital signs recorded during the first 48 hours of admission and the missing ratio is $80.4\%$. We use MSE to measure forecasting performance, with lower values indicating better performance.

**Main Results of Forecasting.** We compare the performance with the ISMTS forecasting models: Graph-based method Grafiti (Yalavarthi et al., 2024), ODE- and RNN-based models including GRU-ODE-Bayes (De Brouwer et al., 2019), Neural Flows (Biloš et al., 2021), CRU (Schirmer et al., 2022), NeuralODE-VAE(Chen et al., 2018), GRUSimple, GRU-D and TLSTM(Baytas et al., 2017). Additionally, attention-based models like mTAND, also an interpolation model, together with variants of Informer (Zhou et al., 2021), Fedformer (Zhou et al., 2022), DLinear, and NLinear (Zeng et al., 2023), denoted as Informer+, Fedformer+, DLinear+, and NLinear+, respectively.

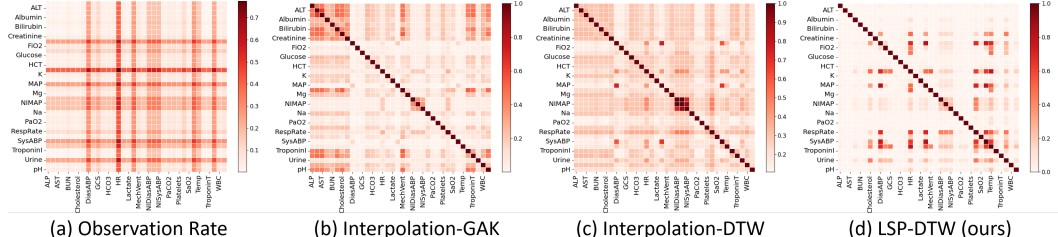

(a) Observation Rate      (b) Interpolation-GAK      (c) Interpolation-DTW      (d) LSP-DTW (ours)

Figure 3: Visualization of various methods to extract the correlation matrix from P12 dataset. The darker the color, the more similar the relationship. (a) denotes the average pairwise observation rate (i.e., 1 minus missing rate), and (b) - (d) denotes different correlation matrices.

Table 4: Classification performance of MuSiCNetr to verify the necessity of correlation matrix.

| Methods | P12 | | P19 | | PAM | | | |
|---|---|---|---|---|---|---|---|---|
| | AUROC | AUPRC | AUROC | AUPRC | Accuracy | Precision | Recall | F1 score |
| **MuSiCNet** | $\mathbf{86.1}\pm_{0.4}$ | $\mathbf{54.1}\pm_{2.2}$ | $\mathbf{86.8}\pm_{0.4}$ | $\mathbf{54.1}\pm_{2.2}$ | $\mathbf{96.3}\pm_{0.7}$ | $\mathbf{96.9}\pm_{0.6}$ | $\mathbf{96.9}\pm_{0.5}$ | $\mathbf{96.8}\pm_{0.5}$ |
| w/o Corr | $85.5\pm_{0.3}$ | $53.0\pm_{2.1}$ | $82.9\pm_{0.8}$ | $32.7\pm_{2.1}$ | $95.7\pm_{0.9}$ | $96.2\pm_{0.51}$ | $96.5\pm_{0.2}$ | $96.3\pm_{0.3}$ |
| Learnable Corr | $85.7\pm_{0.4}$ | $53.0\pm_{2.0}$ | $83.4\pm_{0.7}$ | $31.8\pm_{2.7}$ | $96.1\pm_{0.5}$ | $96.7\pm_{0.38}$ | $96.5\pm_{0.7}$ | $96.6\pm_{0.5}$ |

This experiment is conducted following the setting of GraFITi where for the USHCN dataset, the model observes for the first 3 years and forecasts the next 3 time steps and for other datasets, the model observes the first 36 hours in the series and predicts the next 3 time steps.

As shown in Table 3, MuSiCNet consistently achieves competitive performance across all datasets, maintaining accuracy within the top two among baseline models. While GraFITi excels by explicitly modeling the relationship between observation and prediction points, making it superior in certain scenarios, MuSiCNet remains competitive without imposing priors for any specific task.

### 4.4 CORRELATION RESULTS

In this section, we focus on validating the necessity, effectiveness, and efficiency of the correlation matrix in the classification task as an example.

First, we verify the necessity of the correlation matrix using results from all classification datasets in Table 4. Removing the correlation matrix (line 4) led to performance drops across all datasets, with P19 showing the largest decline due to its 94.9% missing rate. This highlights the importance of capturing inter-series relationships in irregularly sampled time series, making the correlation matrix essential. Replacing the designed correlation matrix with a learnable one (line 5) also worsened performance, indicating that learning inter-series relationships purely from the network remains highly challenging and specialized correlation designs are needed.

Second, we evaluate LSP-DTW against other correlation calculation methods (I-GAK (Cuturi, 2011), I-DTW (Berndt & Clifford, 1994)) on the P12 dataset to verify the effectiveness. Interpolation-based methods (I-GAK, I-DTW) distort correlations, leading to unreliable results as seen in Fig.3. I-GAK shows fictitious correlations based on observation rates, while I-DTW presents uniformly positive correlations, neither of which captures true data characteristics. In contrast, LSP-DTW accurately identifies correlations, verified by Table 5, where it outperforms all baselines, demonstrating the importance of appropriate correlation modeling.

Lastly, we report the computation time for the correlation matrix. LSP-DTW based correlation matrix is computed per instance in parallel, with acceptable runtimes (0.137s for P12, 0.127s for P19, 0.049s for PAM). It is calculated once, making it efficient for the entire learning process.

Table 5: Classification performance of MuSiC-Net with different correlation matrices on P12 to verify the effectiveness.

| Corr Matrix | AUCROC | AUPRC |
|---|---|---|
| Ones | $66.7 \pm 2.2$ | $25.2 \pm 0.3$ |
| Rand | $84.7 \pm 0.8$ | $52.2 \pm 3.2$ |
| Diag | $84.2 \pm 0.8$ | $48.2 \pm 3.4$ |
| I-GAK | $\underline{85.1} \pm 0.6$ | $\underline{52.8} \pm 3.0$ |
| I-DTW | $81.9 \pm 0.6$ | $46.9 \pm 3.0$ |
| LSP-DTW | $\mathbf{86.1} \pm 0.4$ | $\mathbf{54.1} \pm 2.2$ |

Table 6: Ablation studies on different strategies of MuSiCNet in classification. ✓(×) indicates the component has (not) been applied.

| Component | | | P12 | |
|---|---|---|---|---|
| Corr Matrix | Adjustment | Contrastive | AUROC | AUPRC |
| ✓ | ✓ | ✓ | $\mathbf{86.1} \pm 0.4$ | $\mathbf{54.1} \pm 2.2$ |
| × | ✓ | ✓ | $\underline{85.5} \pm 0.3$ | $\underline{53.0} \pm 2.1$ |
| ✓ | × | × | $85.2 \pm 0.6$ | $52.6 \pm 2.5$ |
| ✓ | × | ✓ | $85.4 \pm 0.4$ | $53.0 \pm 2.5$ |
| ✓ | ✓ | × | $85.4 \pm 0.6$ | $52.9 \pm 2.8$ |
| × | × | × | $84.2 \pm 0.8$ | $48.2 \pm 3.4$ |

## 4.5 ABLATION ANALYSIS AND EFFICIENCY EVALUATION

Taking P12 in the classification task with a batch size of 50 as an example, we conduct the ablation study to assess the necessity of two fundamental components of MuSiCNet: correlation matrix and multi-scale learning reflected in reconstruction results adjustment and contrastive learning. As shown in Table 6, the complete MuSiCNet framework, incorporating all components, achieves the best performance. The absence of any component leads to varying degrees of performance degradation, as evidenced in layers two to five. The second layer, which retains the multi-scale learning, exhibits the second-best performance, underscoring the critical role of multi-scale learning in capturing varied temporal dependencies and enhancing feature extraction. Conversely, the version lacking all components shows a significant performance drop of $1.9\%$, indicating that each component is crucial to the overall effectiveness of the framework.

Under the same setting, our model MuSiCNet achieves a time cost of 0.240s per batch with 4.2GB of memory usage. In comparison, ViTST requires 2.196s and 40.2GB, Raindrop uses 0.124s and 4.8GB, MTGNN takes 0.1967s and 4.2GB, and DGM$^2$-O needs 0.313s and 9.1GB. MuSiCNet demonstrates lower time complexity than most other methods and significantly lower memory usage, particularly compared to ViTST, which also performs well on classification tasks.

## 5 CONCLUSION

In this study, we introduce MuSiCNet, an innovative framework designed for analyzing ISMTS datasets. MuSiCNet addresses the challenges arising from data irregularities and shows superior performance in both supervised and unsupervised tasks. We recognize that irregularities in ISMTS are inherently relative and accordingly implement multi-scale learning, a vital element of our framework. In this multi-scale approach, the contribution of extra coarse-grained relatively regular series is important, providing comprehensive temporal insights that facilitate the analysis of finer-grained series. As another key component of MuSiCNet, CorrNet is engineered to aggregate temporal information effectively, employing time embeddings and correlation matrix calculating from both intra- and inter-series perspectives, in which we employ LSP-DTW to develop frequency correlation matrices that not only reduce the burden for similarity calculation for ISMT, but also significantly enhance inter-series information extraction.

Nevertheless, our MuSiCNet still has some limitations. Firstly, the interaction between scales could potentially be further simplified. Secondly, the exploration of ISMTS for anomaly detection tasks is currently insufficient. As a task-agnostic model, our MuSiCNet should be further investigated for its potential in anomaly detection.

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

## A    APPENDIX

## B    PSEUDO CODE FOR MUSICNET

The Pseudo Code is provided using classification as an example. The interpolation task can be obtained by removing the projection head $f_{cls}$ and the classification loss term $\mathcal{L}_{cls}$ from the total loss in line #17. While in the case of forecasting tasks, the projection head will be replaced with $f_{fore}$ and task loss will be changed to $\mathcal{L}_{fore}$ as in Eq.11.

## C    TIME EMBEDDING IN CORRNET

Time Embedding method embeds continuous time points of ISMTS into a vector space Kazemi et al. (2019); Shukla & Marlin (2021). It leverages $H$ embedding functions $\phi_h(t)$ simultaneously and each outputting a representation of size $d_r$. Dimension $i$ of embedding $h$ is defined as follows:

$$\phi_h(t)[i] = \begin{cases} \omega_{0h} \cdot t + \alpha_{0h}, & \text{if} \quad i = 0 \\ \sin\left(\omega_{ih} \cdot t + \alpha_{ih}\right), & \text{if} \quad 0 < i < d_r \end{cases} \tag{7}$$

where the $\omega_{ih}$'s and $\alpha_{ih}$'s are learnable parameters that represent the frequency and phase of the sine function. This time embedding method can capture both non-periodic and periodic patterns with linear and periodic terms, respectively. c

## D    ISMTS ANALYSIS TASKS

The overall loss is defined as Eq.8, incorporating an optional task-specific loss component.

$$\mathcal{L} = \sum_{l=1}^{L} \ell_{recon}^{(l)} + \lambda_1 \ell_{adj}^{(l)} + \lambda_2 \ell_{cons}^{(l)} \tag{8}$$

---

**Algorithm 1** MuSiCNet Algorithm for Classification

---

**Input:** Training set $\mathcal{X}$, the number of scale layers $L$, random masking ratio $r$, max reference point number $|\boldsymbol{\tau}^{(L)}|$, hyper-parameters $\lambda_1$, $\lambda_2$, $\lambda_3$.
**Parameters:** Encoder model $f_{\text{CorrE}}$, decoder model $f_{\text{CorrD}}$, GRU model $f_{\text{GRU}}$, projection head $f_{\text{cls}}$
**Output:** Encoder model $f_{\text{CorrE}}$, GRU model $f_{\text{GRU}}$, projection head $f_{\text{cls}}$

1:   $C_T \leftarrow$ Eq.2 with $\mathcal{X}$
2:   **for** $\boldsymbol{X}$ in $\mathcal{X}$ **do**
3:      $\left\{\boldsymbol{X}^{(1)}, \cdots, \boldsymbol{X}^{(L)}\right\} \leftarrow \text{Mask}_r\left(\text{AvgPooling}_L\left(\boldsymbol{X}\right)\right)$
4:      $\ell_{\text{recon}} \leftarrow 0$
5:      **for** $l \leftarrow 1$ to $L$ **do**
6:         $\boldsymbol{r}^{(l)} \leftarrow f_{\text{CorrE}}\left(X^{(L)}, C_T, |\boldsymbol{\tau}^{(L)}|/2^{(L-l)}\right)$
7:         $\boldsymbol{h}^{(l)} \leftarrow f_{\text{GRU}}\left(\boldsymbol{r}^{(l)}\right)$
8:         $\hat{\boldsymbol{X}}_{\text{recon}}^{(l)} \leftarrow f_{\text{CorrD}}\left(\boldsymbol{r}^{(l)}, |\boldsymbol{X}^{(l)}|\right)$
9:         $\ell_{\text{recon}} \leftarrow \ell_{\text{recon}} +$ Eq.3 with $\boldsymbol{X}^{(l)}$ and $\hat{\boldsymbol{X}}^{(l)}$
10:     **end for**
11:     $\ell_{\text{adj}}, \ell_{\text{cons}} \leftarrow 0, 0$
12:     **for** $l \leftarrow 2$ to $L$ **do**
13:        $\ell_{\text{adj}} \leftarrow \ell_{\text{adj}} +$ Eq.5 with $\hat{\boldsymbol{X}}^{(l-1)}$ and $\hat{\boldsymbol{X}}^{(l)}$
14:        $\ell_{\text{cons}} \leftarrow \ell_{\text{cons}} +$ Eq.6 with $\boldsymbol{h}^{(l-1)}$ and $\boldsymbol{h}^{(l)}$
15:     **end for**
16:     $\mathcal{L}_{\text{cls}} \leftarrow$ Eq.9 with $\boldsymbol{h}^{(L)}$
17:     $\mathcal{L}_{\text{overall}} \leftarrow \frac{1}{L}\ell_{\text{recon}} + \frac{\lambda_1}{L-1}\ell_{\text{adj}} + \frac{\lambda_2}{L-1}\ell_{\text{cons}} + \lambda_3\mathcal{L}_{\text{cls}}$
18:     Update overall network parameters
19: **end for**

---

**Supervised Learning.** We augment the encoder-decoder CorrNet by integrating a supervised learning component that utilizes the latent representations for feature extraction. In this work, we specifically concentrate on classification tasks as a representative example of supervised learning. The loss function is

$$\mathcal{L}_{\text{cls}} = \frac{1}{C}\sum_{c=1}^{C}\frac{1}{n^c}\sum_{i=1}^{n^c}\ell_{CE}\left(\text{CLS}\left(\boldsymbol{h}_i^{(L)}\right), y_i\right) \tag{9}$$

where $C$ denotes the number of classes, $n^c$ denotes the number of samples in $c$-th class, $\text{CLS}\left(\cdot\right)$ denotes the projection head for classification, and $\ell_{CE}\left(\cdot\right)$ denotes the cross-entropy loss.

**Unsupervised Learning.** For our unsupervised learning example, we choose interpolation and forecasting. The loss function for interpolation is defined as

$$\mathcal{L}_{int} = \sum_{n=1}^{N}\left\|\boldsymbol{M}^{(L)}\odot\left((\hat{\boldsymbol{X}}_{\text{reco}}^{(L)})_n - \boldsymbol{X}_n^{(L)}\right)\right\|_2^2 \tag{10}$$

This equation essentially represents the reconstruction outcome at the finest scale as $\ell_{\text{adj}}^{(L)}$ in Eq.4 making the interpolation task fit seamlessly into our model with minimal modifications. Therefore, it is unnecessary to incorporate an additional loss function into our overall loss function Eq.8.

While the loss function for forecasting is defined as

$$\mathcal{L}_{\text{fore}} = \sum_{n=1}^{N}\left\|(\boldsymbol{M}_{\text{fore}})_n\odot\left((\hat{\boldsymbol{X}}_{\text{fore}}^{(L)})_n - (\boldsymbol{X}_{\text{fore}})_n\right)\right\|_2^2 \tag{11}$$

As observations might be missing also in the groundtruth data, to measure forecasting accuracy we average an element-wise loss function $\mathcal{L}_{\text{fore}}$ over only valid values using $(\boldsymbol{M}_{\text{fore}})_n$.

Table 7: Statistics of the ISMTS datasets used in our experiments. "#Avg. obs." denotes the average number of observations for each sample.

| Tasks | Datasets | #Samples | #Variables | #Avg. obs. | #Classes | Imbalanced | Missing ratio |
|---|---|---|---|---|---|---|---|
| Classification | P19 | 38,803 | 34 | 401 | 2 | True | 94.9% |
| | P12 | 11,988 | 36 | 233 | 2 | True | 88.4% |
| | PAM | 5,333 | 17 | 4,048 | 8 | False | 60.0% |
| Interpolation | PhysioNet | 4,000 | 37 | 2,880 | - | - | 78.0% |
| Forecasting | USHCN | 1,100 | 5 | 263 | - | - | 77.9% |
| | MIMIC-III | 21,000 | 96 | 274 | - | - | 94.2% |
| | Physionet12 | 5,333 | 37 | 130 | - | - | 85.7% |

## E  FURTHER DETAILS ON DATASETS

We adopt the data processing approach used in RAINDROP Zhang et al. (2021b) for the classification task, mTANs Shukla & Marlin (2021) for the interpolation task, and GraFITi Yalavarthi et al. (2024) for the forecasting task. The aforementioned processing methods serve as the usual setup, which our method also follows for fair comparison. *However, it's important to note that we do not incorporate static attribute vectors* (such as age, gender, time from hospital to ICU admission, ICU type, and length of stay in ICU) in our processing. This decision is based on the fact that our model, MuSiCNet, is not specifically designed for clinical datasets. Instead, it is designed as a versatile, general model capable of handling various types of datasets, which may not always include such static vectors. The detailed information of baselines is in Table 7.

### E.1  DATASETS FOR CLASSIFICATION

**P19: PhysioNet Sepsis Early Prediction Challenge 2019.**   P19 dataset Reyna et al. (2020) comprises data from $38,803$ patients, each monitored by $34$ irregularly sampled sensors, including 8 vital signs and 26 laboratory values. The original dataset contained $40,336$ patients, but we excluded those with excessively short or long time series, resulting in a range of 1 to 60 observations per patient as in RAINDROP. Each patient has a binary label representing the occurrence of sepsis within the next 6 hours. The dataset has a high imbalance with approximately $\sim 4\%$ positive samples.

**P12: PhysioNet Mortality Prediction Challenge 2012.**   P12 Goldberger et al. (2000) includes data from $11,988$ patients after removing inappropriate 12 samples as explained in Horn et al. (2020). This dataset features multivariate time series from 36 sensors collected during the first 48 hours of ICU stay. Each patient has a binary label indicating the length of stay in the ICU, in which a negative label for stays under 3 days and a positive label for longer stays. P12 is imbalanced with $\sim 93\%$ positive samples.

**PAM: PAMAP2 Physical Activity Monitoring.**   PAM Reiss & Stricker (2012) records the daily activities of 9 subjects using 3 inertial measurement units. RAINDROP has adapted it for irregularly sampled time series classification by excluding the ninth subject for short sensor data length. The continuous signals were segmented into samples with the window size 600 and $50\%$ overlapping rate. Originally with $18$ activities, we retain 8 with over $500$ samples each, while others are dropped. After modification, PAM includes $5,333$ sensory signal segments, each with $600$ observations from $17$ sensors at $100$ Hz. To simulate irregularity, $60\%$ of observations are randomly removed by RAINDROP, uniformly across all experimental setups for fair comparison. The 8 classes of PAM represent different daily activities, with no static attributes and roughly balanced distribution.

### E.2  DATASET FOR INTERPOLATION

**Physionet: PhysioNet Challenge 2012 dataset**   Physionet Reiss & Stricker (2012) comprises 37 variables from ICU patient records, with each record containing data from the first 48 hours after admission to ICU. Aligning with the methodology of Neural ODE Rubanova et al. (2019), we round observation times to the nearest minute, resulting in up to $2,880$ potential measurement times for each time series. The dataset encompasses $4,000$ labeled instances and an equal number of unlabeled instances. For our study, we utilize all $8,000$ instances in interpolation experiments. Our primary

objective is to predict in-hospital mortality, with $13.8\%$ of the instances belonging to the positive class.

### E.3 DATASET FOR FORECASTING

**USHCN: U.S. Historical Climatology Network.**    USHCN Menne et al. (2015) data are used to quantify national and regional-scale temperature changes in the contiguous United States. It contains measurements of 5 variables from 1280 weather stations. Following the preprocessing proposed by De Brouwer et al. (2019), the majority of the over 150 years of observations are excluded, and only data from the years 1996 to 2000 are used in the experiments. Furthermore, to create a sparse dataset, only a randomly sampled $5\%$ of the measurements are retained.

**Physionet12.**    This dataset consists of medical records from $12,000$ ICU patients. During the first 48 hours of admission, measurements of 37 vital signs were recorded. Following the forecasting approach used in recent work, such as Yalavarthi et al. (2024); Biloš et al. (2021); De Brouwer et al. (2019), we pre-process the dataset to create hourly observations, resulting in a maximum of 48 observations per series.

**MIMIC-III: Medical Information Mart for Intensive Care.**    MIMIC-III Johnson et al. (2016) is a widely utilized medical dataset offering valuable insights into ICU patient care. To capture a diverse range of patient characteristics and medical conditions, 96 variables are meticulously observed and documented. For consistency, we followed the preprocessing steps outlined in previous studiesYalavarthi et al. (2024); Schirmer et al. (2022); Biloš et al. (2021); De Brouwer et al. (2019). Specifically, we rounded the recorded observations to 30-minute intervals and used only the data from the first 48 hours post-admission. Patients who spent less than 48 hours in the ICU were excluded from the analysis.

## F    EXPERIMENTAL DETAILS

### F.1    MUSICNET PARAMETERS

We present the training hyperparameters and model parameters here. The maximum epoch is set to 300, and AdamW optimizer is selected as our optimizer without weight decay. By default, the learning rate is set to $1e$-3, and the learning rate schedule is cosine decay for each epoch. Batch size for all datasets is set to 50, the dimension of the encoder output is set to 256, and the dimension of the hidden representations in GRU is typically set to 50. The random masking ratio $r$ for each scale is set to 0.1.

Due to inconsistent series lengths, we set the maximum reference point number, $K,$ to 128 for long series, such as P12, PAM, PhysioNet and USHCN, to 96 for Physionet12, and to 48 for short series, such as PAM and MIMIC-III.

Initially, the window size is set to $1/4$ of the time series length and then halved iteratively until the majority of the windows contain at least one observation.

According to the observed timestamps on each dataset, the number of scale layers $L$ is set to 6, 5, 7, 6, 8, 4, and 5 for P12, P19, PAM, Physionet, USHCN, MIMIC-III, and Physionet12, respectively. For example, in classification, for P12, the scales are $4, 8, 16, 32, 64$ and raw length. For P19, the scales are $4, 8, 16, 32$ and raw length. And for PAM, the scales are $4, 8, 16, 32, 64, 128$ and raw length. In all mainstream tasks involved, the hyperparametes $\lambda_1, \lambda_2, \lambda_3$ are selected in $[1e$-$3, 1e$-$2, \ldots, 1e2]$. All the models were experimented using the PyTorch library on a GeForce RTX-2080 Ti GPU.

### F.2    BASELINE PARAMETERS

The implementation of baseline models adheres closely to the methodologies outlined in their respective papers, including SeFT Horn et al. (2020), GRU-D Che et al. (2018), mTAND Shukla & Marlin (2021) and ViTST Li et al. (2023). We follow the settings of the attention embedding module baseline in mTAND and implement the Multi-Correlation Attention module in our work.

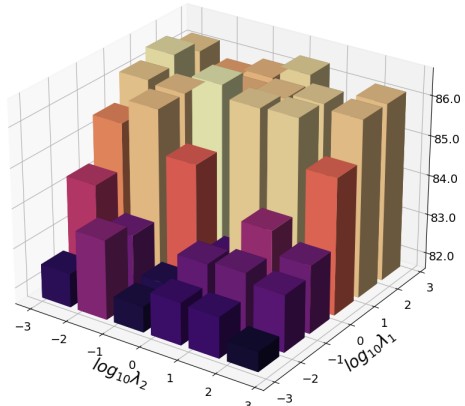 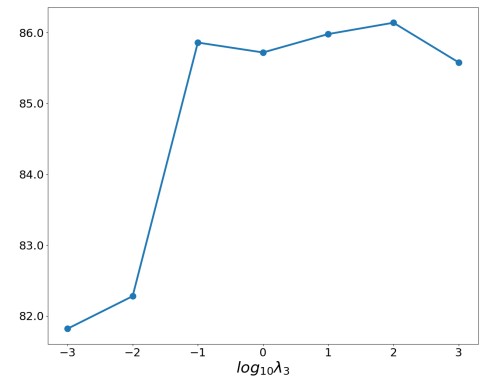

Figure 4: AUCROC performance with varying combinations of hyper-parameter of the adjustment term $\lambda_1$ and hyper-parameter of the contrastive learning term $\lambda_2$ in the logarithmic form on P12

Figure 5: AUCROC performance with varying hyper-parameter of the downstream task $\lambda_3$ in the logarithmic form on P12

### F.3 PARAMETER ANALYSIS

To analyze the hyper-parameters sensitivity, we conducted the experiments for $\lambda_1$, $\lambda_2$, and $\lambda_3$ with grid search. Due to the closer relationship between the hyper-parameters of the adjustment term and the contrastive learning term, i.e., $\lambda_1$ and $\lambda_2$, we jointly analyzed $\lambda_1$ and $\lambda_2$ while separately analyzing the hyper-parameter of the downstream task $\lambda_3$, as illustrated in Fig.4 and Fig.5.

From Fig.4, we can find that the adjustment term plays a greater role compared to the contrastive learning term. This phenomenon matches our motivation, where the coarse-to-fine strategy can effectively alleviate the difficulty of representation learning on ISMTS caused by inconsistent sampling rates. In addition, when $\lg \lambda_1$ and $\lg \lambda_2$ take values around 2 and -2, respectively, our MuSiCNet can perform well.

From Fig.5, we can find that our MuSiCNet becomes effective with large $\lambda_3$. This indicates that more effective representations will be captured when utilizing downstream tasks, matching the general insight. We also noticed that it becomes less sensitive when $\lg \lambda_3 \geq -1$. Its suitable range may be located at $[1e1, 1e2]$.