# OpenReview forum: "MuSiCNet: A Gradual Coarse-to-Fine Framework for Irregularly Sampled Multivariate Time Series Analysis"
_ICLR.cc/2025/Conference — ICLR 2025 Conference Withdrawn Submission_

### Official Review · Reviewer_pftL · 2024-10-30

**Soundness:** 2
**Presentation:** 1
**Contribution:** 2
**Rating:** 3
**Confidence:** 3

**Summary:**

This work focuses on the challenges arising from data irregularities and introduce an innovative framework MuSiCNet designed for analyzing ISMTS datasets.  Through comprehensive experiment discussions, the manuscript demonstrates the effectiveness of the proposed approach beyond existing works.

**Strengths:**

S1. The research field  is meaningful, and the research motivation of this work is clear.

S2. The  experiment discussion is comprehensive.

**Weaknesses:**

W1. I am confused by the insight of irregular relativity. From the way it is presented in the introduction, I find it difficult to see this as a novel insight. And I remain skeptical about whether this insight can effectively guide the resolution of existing challenges.

W2. The definition of the symbols is poorly articulated. What does "with the length of observation $ T_n$" mean?  What do $ T_n$ and $ n$represent, respectively?

W3. What are the dimensions of the variables $ Q $, $ K$, $ A $, and $ C$ in Equation 1? They are not defined before their use.

W4. The writing in the manuscript is quite rough. While I don't deny that this may be an interesting work, it definitely needs substantial revision in terms of writing style. I argue it's crucial to clarify the details and enhance the coupling between each module.

**Questions:**

Q1. What are the generation principles for the hierarchical set? What advantages does this generation mechanism offer?

Q2. On the P19 dataset, what are the reasons that the proposed framework does not stand out?

A typo: Line 497, "IISMTS".

---

> ### Author Response · Authors · 2024-11-20
>
> Thank you for your careful review and helpful suggestions. We note your main concerns about the writing clarity and the insight on irregular relativity. In response, we have made edits to improve readability and further clarified our insights. At the same time, other reviewers have recognized our writing and insights, with comments such as *“The paper is clearly written and well-structured”* (Reviewer ewtP), *“This is a high-quality paper”* (Reviewer T9q8), and *“The paper generally explains its methodology well, with clear explanations of the key concepts and results.”* (Reviewer KxEq). We hope you will also see these strengths in our work. Below is our detailed response:
>
> ### **W1:** The insight of irregular relativity. & **Q1:** The generation principles for the hierarchical set and its advantages.
>
> **A1:** Here, we address W1 and Q1 together.
>
> We rethink the causes of multivariate irregular time series generation and realized that, due to various external forces or interventions mentioned in the paper’s real-world scenarios, sensors are unable to sample uniformly, resulting in ISMTS data.
>
> However, when we zoom in on the time scale, as shown in Figure 1 of the paper, larger time windows are more likely to contain real sampling values, forming relatively regular series, which helps mitigate the issue of irregular sampling because regularly sampled time series have been widely studied and demonstrate to be easier for models to learn.
>
> The advantage of coarse-grained regular sequences is that they ease the learning difficulty caused by irregular sampling as shown above and provide broad-view temporal information. However, fine-grained details that may be missed at this level still need to be captured from finer-grained series, which is why we adopt a hierarchical learning approach. Moreover, it’s important to emphasize that our hierarchical learning iteratively refines the learned representations rather than simply merging them.
>
> ### **W2:** The definition of the symbols is poorly articulated.
>
> **A2:**  In the revised version of the paper, we have updated our symbol notation, highlighted in blue, including but not limited to the dimensions and representation of symbols.
>
> ### **W3:** The dimensions of the variables Q, K, A, and C.
>
> **A3:** Followed by the definition $X_n$ in PROBLEM FORMULATION subsection, the dimensions of the variables can be formulated as $Q \in \mathbb{R}^{|\tau| \times |\tau|}$, $K \in \mathbb{R}^{|\tau| \times T}$, $A \in \mathbb{R}^{|\tau| \times T}$, and $C \in \mathbb{R}^{D \times D}$, where $|\tau|$ denotes the length of reference points, $T$ denotes the length of observed timestamps, and $D$ denotes the number of observed variables in ISMTS.
>
> ### **Q2:** The reasons that the proposed framework does not stand out on the P19 dataset?
>
> **A4:** According to the **no free lunch** theorem, it is natural that no single method excels universally. While our performance may not be the best on certain datasets, our non-SOTA results remain among the top-tier. Moreover, our model demonstrates superior average performance across the three downstream tasks.
>
> Specifically, as described in **subsection 4.1 Time Series Classification**, “For the P19 dataset, while our performance is competitive, MuSiCNet stands out due to its lower time and space complexity compared to ViTST. ViTST converts 1D time series into 2D images, potentially leading to significant space inefficiencies due to the introduction of extensive blank areas, especially problematic in ISMTS.” and the computational complexity is shown in subsection4.5 Ablation Analysis and Efficiency Evaluation.
>
> Even so, we are continually working to improve the performance of our model further.

---

### Official Review · Reviewer_XcmG · 2024-10-31

**Soundness:** 2
**Presentation:** 1
**Contribution:** 1
**Rating:** 3
**Confidence:** 4

**Summary:**

The paper addresses the problem of representation learning for irregularly sampled multivariate time series, where different individual time-series are not synchronized in terms of their measurement times, making it challenging to apply standard time-series modeling approaches. To address this problem the authors propose a multi-scale hierarchical representation scheme built on a combination of ideas from classical time-series modeling (periodograms, dynamic time-warping) and deep learning (encoder-decoder architectures, attention mechanisms, rectification). They then evaluate their proposed methodology across three different time-series tasks (classification, interpolation, forecasting) and compare the effectiveness of their approach to a variety of baselines.

**Strengths:**

- The problem being addressed is important and challenging and occurs in multiple real-world applications such as medical time-series analysis

- The paper describes extensive experimental investigations of the proposed methodology, comparing it with a variety of baselines, across classification, interpolation, and forecasting tasks.

**Weaknesses:**

Contributions and Potential Impact:
- The approached proposed in the paper seems like a potentially useful engineering contribution, one that may work well for time-series problems with certain characteristics - but it less clear if the paper provides any significant advance in general time-series modeling methodology that would be of interest to the ICLR community and to time-series researchers more broadly.

- A general issue with the paper is that the proposed methodology combines quite a few different heuristic choices, resulting in a  complex overall model (e.g., Figure 2) that involves multiple hyperparameters and design choices. As a consequence, given the types of inductive biases that are being built into the approach (implicitly, via the design choices), it makes sense to wonder what types of problems the proposed methodology will work well, and on what types of problems it will have limitations. The paper would be much stronger if it could be extended to provide this type of insight to the reader: for example, see suggestions below about potentially expanding the discussion of limitations, and possible inclusion of simulations to provide additional insight.



Writing:
- The paper could be improved and be of more value to a reader by improving the writing.

- As an example, there are a few key terms used throughout the paper where it would be of great help to the reader if the terms were more clearly and precisely defined. For the term "ISMTS" its unclear what the scope of this term is: can each of the individual time-series be sampled at arbitrary times, in the general case? or are there any restrictions on this level of generality? You could use Figure 1 to help explain this to the reader: from looking at a blown-up version of Figure 1 (scale L), it would appear that there are no restrictions and that each time-series can be sampled at any arbitrary set of times: is this the case?. As an example of where this is described much more clearly, see Figure 1 in the Yalavarthi et al (2024) paper where the authors clearly illustrate different sampling scenarios and make it clear which one they are focusing on in their paper.

- Another key term that is unclear is "missing ratio" which is used to characterize the datasets in Section 4: how is this defined? does it imply that all of the time-series are potentially being measured on the same discrete time-scale (e.g., hourly or daily) but that some of the time-series don't have measurements at each discrete time (e.g., are only measured every few hours or every few days) and the "missing ratio" is the number of missing measurements relative to this fixed sampling scheme? If so, then this would seem to imply that the datasets used in experiments are a special case of the general framework (since they all have a "missing ratio" defined, Table 7) which has implications for the interpretation of the experimental results and limits the generality of the claims earlier in the paper. Another possibility is that the "missing ratio" has a different interpretation than my guess above. Whichever is the case, it needs to be clearly defined for the reader.

- As another example related to writing, in section 3.2, the key novel contribution of the paper, the CorrNet architecture is introduced. The writing here could be significantly improved by trying to impart more insight to the reader, for example by providing a clear and intuitive toy example (for example with just 3 time-series) of what the method is doing. The current description comes across as somewhat black-box in nature. In particular, while the inclusion of different components (attention, correlation, rectification, etc) each seem sensible from a high-level viewpoint, the reader may wonder how the different parts will work together. For example, it would be helpful to be realistic here and explain when and why these methods should work well, and when we might expect them to fail.

- Figure 2 in its current form will be very difficult for a reader to understand. At a minimum I suggest that you devote part of the text in the paper to a clear description of how data flows through the model to accompany Figure 2 (currently the text mentions Fig 2(a) in one location, and Fig 2(b) in another and I didn't see a reference to Fig 2(c) in the main text, but may have missed it). You may need to change the figure so that you can more clearly describe the steps.  (again, using Yalavarthi et al (2024) as an example, their Figure 3 is a much clearer representation of information flow and their general approach than your Figure 2). Alternatively, skip the figure and use the space to more clearly describe the mapping (input-output) that it represents, with a clear sequence of equations and/or text.


Discussion of Limitations:
- The discussion of limitations of the approach (middle of page 10) is limited in terms of detail and scope. Readers would find it very helpful here to have a more realistic evaluation of the strengths and weaknesses of the approach. For example: what is the sensitivity of the method to architecture design choices? how sensitive are the results to hyperparameter tuning? are there situations where the LSP/DTW approach will fail? And so on.

Insights from Simulations:
- simulation results could also be quite helpful in providing insights into the method. For example, you could simulate datasets from some predefined (known) continuous-time multivariate temporal process and then evaluate, both theoretically and empirically, the effect of different sampling schemes for the time series, potentially making connections to concepts such as Nyquist sampling in this context, and how such concepts might be relevant to your proposed approach.  This would be a valuable addition to the paper and complement the leaderboard-style experiments in Section 4.


Minor suggestion:
- a name other than "MusicNet" is worth considering since the name "MusicNet" will immediately suggest to a reader that this paper is about neural networks for music (rather than the actual more general topic of the paper). And there is a well-known "Musicnet" dataset already in the literature.

**Questions:**

- please clarify definitions of the terms ISMTS and "missing ratio" (see weaknesses above)

- what is meant in  Section 3.2 by "materializing its output ... time points \tau = [ ..]". How is k selected here? what is the sensitivity of the overall approach to the selection of k?

- at lines 245-246 how are the K reference time points defined? is this a different K to the lowercase k above?

- the baseline methods (e.g., for classification at bottom of page 6, and for interpolation at the bottom of page 7) seem a little old, at least on the timescale of advances in deep learning, with most references being from the 2018-2021 period. Are there any more recent baselines that are more SOA and that would be worth comparing to? For example, for the interpolation task in particular it seems like there are quite a few more recent (and more accurate) baseline methods that you could have considered (see for example the list of post-2019 methods in Table 1 of Wang et al (2024) https://arxiv.org/pdf/2402.04059)

- in your forecasting experiments, is there a reason you omitted the LinODEnet method from Scholz et al (2023): in the Yalavarthi et al (2024) paper this was the next-best method to their Grafiti method, so it seems like it would be worth including in your evaluation.

- could one come up with a baseline method based on non-neural architectures, such as some form of a multivariate Gaussian process for example? This might not work particularly well empirically and might require some ad-hoc pre-processing to implement, but some discussion of why and when such an approach would not work well, and/or a demonstration of this in your experiments, would be helpful to the reader. (The non-neural approach need not be based on a Gaussian process, this is just one possible choice of a non-parametric model).

---

> ### Author Response · Authors · 2024-11-20
>
> Thank you for your suggestions. We noted your concerns regarding Writing, the Discussion of Limitations, and Insights from Simulations. Below, we provide specific responses to each of your points. **We also noticed that most of your reviews are *suggestions*, such as adding a hyperparameter sensitivity analysis or including toy data to validate the model’s effectiveness. Therefore, we hope you can improve your evaluation of our paper.** If any questions remain or you feel unable to increase the paper’s score, please feel free to ask, we would be glad to clarify.
>
> Regarding your comments on writing, our responses are as follows:
>
> 1. The term ISMTS, which we use throughout our paper, is common in the field, not coined by us. For example, the paper by Yalavarthi et al. (2024) also uses "irregularly sampled multivariate time series," and this term appears in their Figure 1. Moreover, though we did not use a specific figure to introduce ISMTS,  we define ISMTS and its key characteristics and draw Figture 1 to facilitate readers' understanding in the first paragraph of the introduction.
> 2. In real-world applications, sensors cannot record continuously, so the data collected is necessarily discrete. “Missing ratio” is a common term that describes the percentage of missing observations in a time series (similar to “sparsity” in Yalavarthi et al. (2024), as you noted). Although the “missing ratio” assumes regular sampling, it is widely used to describe ISMTS data characteristics. This term is also explained on the dataset website referenced in our paper and many other related work.
> 3. Our paper focuses on real-world datasets with substantial missing ratios, which are influenced by various external forces or interventions, leading to irregular sampling. Moreover, the effectiveness of our model can be demonstrated by the performance of the downstream tasks performed on datasets with high missing ratios and is unnecessary to use toy data.
> 4. We have clarified in the legend of Figure 2 that gray arrows indicate the direction of data flow and the subsection “Encoder-Decoder Framework” also detailly introduces the whole model processing. Additionally, we corrected the error in the subsection “Correlation Extraction”, where we mistakenly referred to Fig.2(c) as Fig.2(b), and have updated this in the revised version.
>
> Based on the **discussion of limitations,** we  conduct sensitivity analysis experiments in subsection F.3 Parameter Analysis in the appendix as follows
>
> “To analyze the hyper-parameters sensitivity, we conducted the experiments for $\lambda_1$, $\lambda_2$, and $\lambda_3$ with grid search. Due to the closer relationship between the hyper-parameters of the adjustment term and the contrastive learning term, i.e., $\lambda_1$ and $\lambda_2$, we jointly analyzed $\lambda_1$ and $\lambda_2$ while separately analyzing the hyper-parameter of the downstream task $\lambda_3$, as illustrated in Figure 4 and 5.
> From Figure 4, we can find that the adjustment term plays a greater role compared to the contrastive learning term.
> This phenomenon matches our motivation, where the coarse-to-fine strategy can effectively alleviate the difficulty of representation learning on ISMTS caused by inconsistent sampling rates. In addition, when $\lg \lambda_1$ and $\lg \lambda_2$ take values around 2 and -2, respectively, our MuSiCNet can perform well.
> From Figure 5, we can find that our MuSiCNet becomes effective with large $\lambda_3$.
> This indicates that more effective representations will be captured when utilizing downstream tasks, matching the general insight.We also noticed that it becomes less sensitive when $\lg \lambda_3 \ge -1$. Its suitable range may be located at $\left[1e1, 1e2\right]$.“
>
> Regarding the question on **Insights from Simulations**, the goal of our paper is to learn effective representations of ISMTS, which can be validated through good performance on downstream tasks, demonstrating the quality of the learned representations. Additional toy data is not necessary to illustrate this point. Additionally, the frequently mentioned Yalavarthi et al. (2024) paper also does not employ simulation experiments.
>
> According to the **minor suggestion**, we have changed the paper title to “A Gradual Coarse-to-Fine Framework for Irregularly Sampled Multivariate Time Series Analysis.”

---

> > ### Author Response · Authors · 2024-11-20
> > **Official Comment by Authors_2**
> >
> > ### **Q2**: what is meant in Section 3.2 by "materializing its output ... time points \tau = [ ..]". How is k selected here? what is the sensitivity of the overall approach to the selection of k?
> >
> > **A2**:
> >
> > We aims to learn fixed-length representations from ISMTS data, with the number of reference points, $K$, representing the chosen length of the learned representation.
> >
> > The selection of  $K$   in this paper is based on series length, with specific values provided in Appendix subsection F.1 MuSiCNet parameters, where we state: *“Due to inconsistent series lengths, we set the maximum reference point number,* $K$ *, to 128 for long series, such as P12, PAM, PhysioNet and USHCN, to 96  for PhysioNet12, and to 48 for short series, such as PAM and MIMIC-III.”* Therefore, we did not adjust $K$  in the experiments, but instead pre-set it accordingly.
> >
> > ### **Q3**: at lines 245-246 how are the K reference time points defined? is this a different K to the lowercase k above?
> >
> > **A3**: The selection of  $K$   in this paper is based on series length as in the above answer. These two  $K$  actually refer to the same parameter. We mistakenly used lowercase  $k$ instead of uppercase  $K$, and we have corrected this issue in the latest revision.
> >
> > ### **Q4**: the baseline methods seem a little old, at least on the timescale of advances in deep learning.
> >
> > We must emphasize that **imputation and interpolation are two distinct tasks with different settings.**
> >
> > **Interpolation** methods are often used to provide an interface between irregularly sampled time series data, allowing for the estimation of observations at any desired time point.
> >
> > **Imputation** methods, on the other hand, first convert irregularly sampled time series data into a regularly sampled series and then fill in missing values to achieve a consistent sampling rate.
> >
> > We find a recent study [1] and compare our interpolation task results with theirs.
> >
> > | Observed % | 50% | 70% | 90% |
> > | --- | --- | --- | --- |
> > | Model | MSE ($\times 10^{-3}$) | MSE ($\times 10^{-3}$) | MSE ($\times 10^{-3}$) |
> > | NIERT | 2.868±0.021 | 2.656±0.041 | 2.709±0.157 |
> > | NIERT w/pretraining | 2.831±0.021 | 2.641±0.052 | 2.596±0.159 |
> > | MuSiCNet | **0.918 ± 0.025** | **0.938 ± 0.014** | **0.965 ± 0.008** |
> >
> > [1] Ding S, Xia B, Ren M, et al. NIERT: Accurate Numerical Interpolation through Unifying Scattered Data Representations using Transformer Encoder[J]. IEEE Transactions on Knowledge and Data Engineering, **2024**.
> >
> > ### **Q5**: could one come up with a baseline method based on non-neural architectures
> >
> > **A5**: This task falls outside the scope of our study. Given the need for concise and focused writing in a conference paper, we do not consider this requirement essential.

---

> > ### Comment · Reviewer_XcmG · 2024-11-26
> >
> > Thank you for your response. A few followup comments:
> >
> > -  "We also noticed that most of your reviews are suggestions....Therefore, we hope you can improve your evaluation of our paper". I am not sure I understand this comment - the evaluation of the paper is indeed mostly in the form of suggestions for improvement.
> >
> > - although ISMTS may be common among the papers you cite, its not a widely used term in time-series analysis (or indeed among ICLR attendees/readers in general). I encourage you to provide a precise definition early in the paper to make the paper more accessible to a wide audience. I also encourage you to include a simple definition of what you mean by "missing ratio" (again in the interests of being precise and avoiding ambiguity in the reader's mind) rather than asking readers to look this up in other sources.
> >
> > - For simulation experiments, while simulations are not strictly necessary, they could be quite insightful for a reader and can provide a controlled way to explore strengths and weaknesses of a particular modeling approach (in the absence of applicable theory). I would encourage you to consider meaningful simulations that could tease out both strengths and weaknesses of your approach. And the fact that a previous paper doesn't have simulations does not mean it would not be useful.
> >
> > - thank you for correcting the Fig 2 reference and for updating the paper title.

---

> > > ### Comment · Reviewer_XcmG · 2024-11-26
> > >
> > > - thank you for the clarification about k and K.  This seems like an important hyperparameter for the method, one that needs to be selected heuristically by a user. I suggest you to give more attention to discussing the selection of K in the main section of the paper and potentially some discussion of potential sensitivity of the method to its choice.
> > >
> > > - thank you for adding the comparison to NIERT (with the numbers for their method from Table VII in their paper for the Physionet data? and your numbers taken from Table 2 in your paper?). Your numbers show a roughly factor of 3 reduction in MSE over NIERT. If you plan to include these results in the next iteration of your paper it would be helpful to provide the reader with some intuition as where the major reduction in error is coming from. Also, given that there is such a large reduction, it would be good to double-check the results, for example by independently replicating their results by directly running their code and your code on the same datasets.
> > >
> > > - any clarification about why the LinODEnet method was not included in your experiments?
> > >
> > > - I disagree that non-neural methods are "outside the scope." There is a large literature on methods for non-parametric curve fitting and for interpolation in general and it would be of great benefit to a reader to show how your approach (and the other baselines) compare to some standard non-neural approaches. You would not need to spend much time on them in the main paper, just include a few in your results, with more details relegated to the Appendix.

---

### Official Review · Reviewer_KxEq · 2024-11-03

**Soundness:** 3
**Presentation:** 2
**Contribution:** 3
**Rating:** 6
**Confidence:** 4

**Summary:**

This paper introduces MuSiCNet, a novel framework for analyzing irregularly sampled multivariate time series (ISMTS). MuSiCNet addresses the challenges posed by ISMTS data by employing a multi-scale approach and a custom-designed encoder-decoder framework called CorrNet. The multi-scale approach allows the model to learn from both coarse-grained and fine-grained representations of the data, capturing long-term dependencies and detailed temporal variations. CorrNet utilizes time attention and frequency correlation matrices to aggregate information within and across series, improving the quality of the learned representations. The paper demonstrates the effectiveness of MuSiCNet on three mainstream ISMTS tasks: classification, interpolation, and forecasting, achieving competitive performance compared to state-of-the-art methods.

**Strengths:**

S1: The multi-scale approach and the use of frequency correlation matrices for inter-series attention are novel contributions to ISMTS analysis.

S2: The framework is experimentally validated with strong results on multiple benchmarks, indicating reliability. The results demonstrate the effectiveness of the proposed method.

S3: The paper generally explains its methodology well, with clear explanations of the key concepts and results.

S4: This work addresses a broad need in ISMTS analysis, particularly in medical and environmental fields, where irregular data sampling is common.

**Weaknesses:**

W1. The paper's summary and comments on current research work are not accurate and rigorous enough. For example:

(1) "Most existing methods treat ISMTS as synchronized regularly sampled time series with missing values." However, many recent studies do not treat ISMTS as synchronized but as asynchronous, such as in the papers "Set Functions for Time Series" and "Graph-Guided Network for Irregularly Sampled Multivariate Time Series."

(2) "Neglecting that the irregularities are primarily attributed to variations in sampling rates." However, "A review of irregular time series data handling with gated recurrent neural networks" has already pointed out and explained that different sampling frequencies lead to irregular time series data.

(3) "They rely on assumptions tailored to specific downstream tasks, hindering their ability to consistently perform well across various ISMTS tasks," yet the paper does not elaborate on or provide examples of these "assumptions tailored to specific downstream tasks."

W2. The language used in the paper is not precise enough, with extensive use of "in some senses," "to some extent," which makes it difficult to accurately gauge the degree the authors intend to convey.

W3. The paper lists the model's versatility across tasks as one of its three main contributions, yet many recent models can be applied to multiple tasks. For example, the papers "Latent ODEs for Irregularly-Sampled Time Series," "ContiFormer: Continuous-Time Transformer for Irregular Time Series Modeling," and "IVP-VAE: Modeling EHR Time Series with Initial Value Problem Solvers."

W4. Figure 2 (a) appears to have poor readability. The caption mentions three main components, but their positions and relationships are not clearly displayed in the figure. X appears to be the input, but the final output is not clearly indicated.

W5. The paper states, "MuSiCNet stands out due to its lower time and space complexity compared to ViTST," yet there is no theoretical derivation or experimental results provided in the paper to support this claim.

W6. The paper uses Informer, Fedformer, DLinear, etc., as baselines for time series forecasting, but these models were originally only suitable for regular time series data? How were they adapted and applied to irregular time series data? The paper does not specify.

W7. There is a spelling error in the Conclusion section. "...designed for analyzing IISMTS datasets" where "IISMTS" should probably be "ISMTS."

**Questions:**

1. How to accurately understand “irregularity is essentially relative in some senses”?

2. How sensitive is MuSiCNet to the choice of hyperparameters, such as the number of scales and the masking ratio?

3. How does the computational complexity of MuSiCNet compare to other ISMTS analysis methods?

---

> ### Author Response · Authors · 2024-11-20
>
> Thank you for your positive feedback in the Strengths section. We noticed that your concerns primarily revolve around related work, certain writing approaches, and experimental details. We have addressed these concerns and revised the paper accordingly. We hope our efforts will enhance your recognition of our work, and we would greatly appreciate it if you could consider raising your score for our paper. If you have any further questions, please feel free to ask—we are more than happy to provide additional clarification. Below are our detailed responses.
>
> ### **W1:** The paper's summary and comments on current research work are not accurate and rigorous enough.
>
> **A1:** 1. When we state, "Most existing methods treat ISMTS as synchronized regularly sampled time series with missing values," we use "most" to acknowledge that not all papers take this approach; otherwise, we would have used "all."
> 2. In the sentence, "Neglecting that the irregularities are primarily attributed to variations in sampling rates," this phrase directly follows the previous statement 1 above as part of the same sentence, separated by a comma. Thus, we mean that methods treating ISMTS as synchronized regularly sampled time series with missing values often overlook this aspect. Given that many studies do approach ISMTS in this way, but our own method does not follow this method, we found it necessary to briefly point out this potential limitation in the abstract.
>
> As you mentioned, irregularly sampled time series indeed arises from varying sampling frequencies. However, many existing methods either fail to recognize this or do not leverage this information in their modeling. Our approach aims to effectively utilize this characteristic to better model ISMTS.
>
> 1. Here, we refer to the fact that many existing ISMTS analysis models are designed for specific tasks, such as prediction, classification, or imputation. We have updated our manuscript to include additional references on this. In contrast, our work focuses on learning a good representation that can adapt to multiple downstream tasks.
>
> ### **W2:** The language used in the paper is not precise enough.
>
> **A2:** The phrase "irregularity is essentially relative in some senses" reflects our perspective on ISMTS based on varying sampling rates. Another perspective considers ISMTS as regularly sampled time series with missing values, attributing the irregularities to missing data. Given these differing viewpoints, it would be inaccurate to state outright that irregularities are relative. However, since we already use the term "mitigate," we have decided to remove "to some extent.”
>
> ### **W3:** The paper lists the model's versatility across tasks as one of its three main contributions.
>
> **A3:**  We emphasize here that our primary focus is **on learning a good representation, enabling our model to be not limited to a specific analysis task but to serve as a task-general framework for ISMTS analysis**.
>
> **While some existing works are capable of handling multiple tasks, such cases are relatively rare, and the tasks are addressed differently.** For instance, ContiFormer performs classification and event prediction, IVP-VAE focuses on classification and forecasting, and Latent ODE tackles interpolation, extrapolation, and some dataset-specific tasks.
>
> Therefore, we believe that achieving multiple tasks remains one of the notable contributions of our work at this stage.
>
> ### **W4**: Figure 2 (a) appears to have poor readability.
>
> **A4**:  (a) represents the overall framework of our model. Based on the title of (b), we can see that it corresponds to the **CorrNet Encoder** in (a). (c) illustrates the **Frequency Correlation Matrix**, which is the method used in (b) to compute the inter-series relationships in the **Correlation Matrix**.
>
> $X$ denotes the input, and the representation $r^{(L)}$ learned by the final layer,  $L$, of the **CorrNet** is considered the model's output, as all subsequent downstream tasks are performed on $r^{(L)}$.

---

> ### Author Response · Authors · 2024-11-20
> **Official Comment by Authors_2**
>
> ### **W5:** The paper states, "MuSiCNet stands out due to its lower time and space complexity compared to ViTST," yet there is no theoretical derivation or experimental results provided in the paper to support this claim.
>
> **A5:** We conduct experiments on the P12 dataset with a batch size of 50. Empirically, our model MuSiCNet requires 0.240s per batch with 4.2GB memory usage, compared to ViTST’s 2.196s and 40.2GB, Raindrop’s 0.124s and 4.8GB, MTGNN’s 0.1967s and 4.2GB, and DGM$^2$-O’s 0.313s and 9.1GB. Our model achieves significantly lower time complexity than the other methods and uses much less memory than ViTST, which also performs well on classification tasks. We will add this in section 4.5 “Ablation Analysis and Efficiency Evaluation”.
>
> ### **W6:** How were they adapted and applied to irregular time series data?
>
> **A6:** We share the same setting with [1], so we directly used their results. And they did not mention how they were adapted.
>
> ### **W7:** There is a spelling error in the Conclusion section.
>
> **A7:** We carefully review the entire paper and correct spelling errors.
>
> ---
>
> ### **Q1:** How to accurately understand “irregularity is essentially relative in some senses”?
>
> **A8:** This means that, to some extent, irregular sampling can be transformed into regular sampling. Specifically, as mentioned in our paper: *"With sampling rates artificially determined from low to high, an irregularly sampled time series can be transformed into a hierarchical set of relatively regular time series from coarse to fine."*
>
> ### **Q2:** How sensitive is MuSiCNet to the choice of hyperparameters, such as the number of scales and the masking ratio?
>
> **A9:** In Appendix subsection F.1 MUSICNET PARAMETERS, we explain the selection method for *the number of scales*. This approach does not involve manual selection. According to the observed timestamps in each dataset, we ensure that in the (L-1)-th layer, most windows contain at least one sampling point, while the L-th layer represents the original data layer.
>
> As for the *masking ratio*, we set it uniformly to 10%, as mentioned in subsection F.1. This is because the missing rate in ISMTS is already significantly high. Therefore, we follow the smallest masking ratio in MAE [1], which is 10%, and apply it to all the experiments in this paper.
>
> Consequently, we did not conduct sensitivity analysis for these two parameters. However, we  conduct sensitivity analysis experiments in subsection F.3 Parameter Analysis in the appendix as follows
>
> “To analyze the hyper-parameters sensitivity, we conducted the experiments for $\lambda_1$, $\lambda_2$, and $\lambda_3$ with grid search. Due to the closer relationship between the hyper-parameters of the adjustment term and the contrastive learning term, i.e., $\lambda_1$ and $\lambda_2$, we jointly analyzed $\lambda_1$ and $\lambda_2$ while separately analyzing the hyper-parameter of the downstream task $\lambda_3$, as illustrated in Figure 4 and 5.
> From Figure 4, we can find that the adjustment term plays a greater role compared to the contrastive learning term.
> This phenomenon matches our motivation, where the coarse-to-fine strategy can effectively alleviate the difficulty of representation learning on ISMTS caused by inconsistent sampling rates. In addition, when $\lg \lambda_1$ and $\lg \lambda_2$ take values around 2 and -2, respectively, our MuSiCNet can perform well.
> From Figure 5, we can find that our MuSiCNet becomes effective with large $\lambda_3$.
> This indicates that more effective representations will be captured when utilizing downstream tasks, matching the general insight.We also noticed that it becomes less sensitive when $\lg \lambda_3 \ge -1$. Its suitable range may be located at $\left[1e1, 1e2\right]$.“
>
> [1] He, Kaiming, et al. "Masked autoencoders are scalable vision learners." *Proceedings of the IEEE/CVF conference on computer vision and pattern recognition*. 2022.
>
> ### **Q3:** How does the computational complexity of MuSiCNet compare to other ISMTS analysis methods?
>
> **A10:** Please refer to our response in Weakness 5.

---

> > ### Comment · Reviewer_KxEq · 2024-11-25
> >
> > Thanks for your response. I have no more questions and will maintain my score.

---

### Official Review · Reviewer_T9q8 · 2024-11-03

**Soundness:** 2
**Presentation:** 2
**Contribution:** 2
**Rating:** 5
**Confidence:** 3

**Summary:**

1. Innovation: The paper proposes a novel MuSiCNet framework for the analysis of irregularly sampled multivariate time series, which is innovative. It views irregularity from a new perspective and effectively solves the problems of existing methods through multi-scale and multi-correlation attention mechanisms.
  2. Rationality of the method: The method is reasonably designed. For example, using LSP - DTW to calculate the frequency correlation matrix can effectively handle the correlation problem in irregularly sampled time series and avoid the spurious correlations that may be generated by existing distance measurement methods. The processing methods within and across adjacent scales have theoretical bases and their effectiveness has been verified through experiments.
  3. Adequacy of experiments: The experimental part is very sufficient. In three mainstream tasks of classification, interpolation, and forecasting, multiple real-world datasets (such as P19, P12, PAM, PhysioNet, USHCN, MIMIC - III, etc.) are used to compare with a variety of advanced methods. The results show that MuSiCNet is competitive, verifying the effectiveness and generality of the framework.
  4. Limitations and suggestions: The paper also points out some limitations, such as the interaction between scales may be further simplified and the exploration of anomaly detection tasks is insufficient. It is recommended that the authors further study these problems in future work to improve the performance and applicability of the model.
  5. Overall evaluation: This is a high-quality paper that proposes a promising method for the analysis of irregularly sampled multivariate time series and makes an important contribution to the research in this field. It is recommended to be accepted.

**Strengths:**

- Innovation: The proposed MuSiCNet framework is novel. It offers a new perspective on irregularity and effectively addresses existing method problems via multi-scale and multi-correlation attention mechanisms.
- Rationality of the method: The method is well-designed. LSP - DTW for calculating the frequency correlation matrix is effective in handling correlations and avoiding spurious ones. Processing methods within and across scales have theoretical support and experimental verification.
- Adequacy of experiments: The experiments are comprehensive. Multiple real-world datasets are used in three mainstream tasks, and comparisons with advanced methods show the competitiveness of MuSiCNet,     validating its effectiveness and generality.

**Weaknesses:**

- In the forecasting task performance, the model isn't SOTA(state-of-the-art) in two out of three datasets.
- Moreover, there's no comparison of model complexity, training, and inference costs with baselines, making it hard to evaluate its efficiency.
- Some motivations have not been proved by experiments directly.

**Questions:**

- Performance in FORECASTING task: In the TIME SERIES FORECASTING task, the model is not SOTA on 2 out of 3 datasets. Further analysis is needed. Also, model complexity, training, and inference costs compared to baselines should be provided.
- Multi-scale training details: Regarding the multi-scale learning method, it's unclear if the number of training epochs is the same as without multi-scale training. Also, the relationship between sample numbers at different scales needs clarification.
- Experimental  validation of coarse-grained series: There is a lack of sufficient experimental analysis to prove that the coarse-grained series can help the representation capture broad-view temporal information as claimed. Although an ablation study shows the effectiveness of a related loss term, it doesn't directly prove this crucial aspect, which may affect the method's theoretical and practical credibility.

---

> ### Author Response · Authors · 2024-11-20
>
> We sincerely thank the reviewer for their careful reading and insightful comments. Your feedback primarily focuses on the performance in the forecasting task, the details of multi-scale training, and the experimental validation of coarse-grained series. We have addressed these points and hope our responses resolve your concerns, encouraging you to consider raising your rating of our work. If you have any further questions, please feel free to ask. Below are our detailed responses:
>
> ### **Q1:** In the TIME SERIES FORECASTING task, the model is not SOTA on 2 out of 3 datasets. Further analysis is needed. Also, model complexity, training, and inference costs compared to baselines should be provided.
>
> **A1:**
>
> 1. According to the no free lunch theorem, it is natural that no single method excels universally. While our performance may not be the best on certain datasets, our non-SOTA results remain among the top-tier. Moreover, our model demonstrates superior average performance across the three downstream tasks. Specifically, as described in **subsection 4.3 Time Series Forecasting**, the model achieving SOTA results in the forecasting task is specifically designed for that purpose, incorporating priors designed for forecasting. In contrast, our MuSiCNet remains competitive without relying on task-specific priors. Even so, we are continually working to improve the performance of our model further.
>
> 2.  We conduct experiments on the P12 dataset with a batch size of 50. Empirically, our model MuSiCNet requires 0.240s per batch with 4.2GB memory usage, compared to ViTST’s 2.196s and 40.2GB, Raindrop’s 0.124s and 4.8GB, MTGNN’s 0.1967s and 4.2GB, and DGM$^2$-O’s 0.313s and 9.1GB. Our model achieves significantly lower time complexity than the other methods and uses much less memory than ViTST, which also performs well on classification tasks. We will add this in section 4.5 “Ablation Analysis and Efficiency Evaluation”.
>
> ### **Q2:** Regarding the multi-scale learning method, it's unclear if the number of training epochs is the same as without multi-scale training. Also, the relationship between sample numbers at different scales needs clarification.
>
> **A2: In our setup, the training epochs remain the same and are not affected by the multi-scale design.**
>
> In Appendix subsection F.1 MUSICNET PARAMETERS, we explain the selection method for *the number of scales*. **This approach does not involve manual selection.** According to the observed timestamps in each dataset, we ensure that in the (L-1)-th layer, most windows contain at least one sampling point, while the L-th layer represents the original data layer.
>
> ### **Q3:** Experimental validation of coarse-grained series.
>
> **A3:** From the sensitivity analysis in Figure 5, we can find that compared to $\lambda_2$, the hyperparameter of the adjustment term, i.e., $\lambda_1$, plays a greater role, which aims to utilize broader temporal information (larger scale) to guide the effectiveness of detailed temporal information (smaller scale), to boost the representation learning. If $\lambda_1$ did not become a key role, then our MuSiCNet would definitely lose the guidance of broad-view temporal information. However, this did not happen. Moreover, Scale 1 in Figure 1 clearly illustrates what the broad-view trend information. Therefore, the above phenomenon can indirectly validate that the coarse-grained series can help the representation capture broad-view temporal information.

---

### Official Review · Reviewer_ewtP · 2024-11-04

**Soundness:** 2
**Presentation:** 3
**Contribution:** 3
**Rating:** 5
**Confidence:** 5

**Summary:**

This paper proposes a framework for modeling irregularly sampled multivariate time series (ISMTS). The key contributions include leveraging Lomb-Scargle Periodogram-based Dynamic Time Warping (LSP-DTW) to enhance inter-series correlation modeling and adopting a multi-scale approach to iteratively refine representations across multiple scales. The framework is evaluated across three ISMTS tasks: classification, interpolation, and forecasting.

**Strengths:**

1. Hierarchical modeling for irregularly sampled multivariate time series is a promising and feasible approach.

2. The use of DTW to measure similarity between variables and construct a correlation matrix is intuitively sound and well-motivated.

3. The paper is clearly written and well-structured.

**Weaknesses:**

1. The paper’s core contributions, particularly the hierarchical design and the DTW-based correlation matrix, are not fully demonstrated in the ablation study. Key missing analyses include:

a) Comparing multi-scale modeling with single-scale modeling to quantify the benefits. An assessment of the optimal number of scales, along with the computational cost added by each scale.

b) Evaluating the LSP-DTW correlation matrix against inter-variable self-attention, as seen in prior work (e.g., Warpformer), to confirm its advantage over attention-based correlation modeling.

2. The LSP-DTW computation introduces additional overhead, raising concerns about efficiency. The authors should provide a detailed analysis of the computational trade-offs to demonstrate if the performance gains justify the added complexity.

3. The paper mischaracterizes Warpformer as a multi-scale model for regularly sampled time series, while Warpformer actually pioneered multi-scale modeling for irregular time series using DTW. A detailed comparison with Warpformer, including both similarities and distinct contributions, is essential for clarity. Besides, it should be included as an important baseline to be compared with.


Minor suggestion: Address minor typos, such as the use of \citep{} and \cite{}.

**Questions:**

- Can MuSiCNet be viewed as a hierarchical extension of mTAND, with the addition of a correlation matrix to capture inter-variable similarity?

- The approach to handling irregularity in MuSiCNet closely resembles that of mTAND, aside from the use of DTW. Are there any other unique design elements aimed at better adapting to irregular data?

- What specific challenges does hierarchical modeling pose for irregularly sampled time series compared to regular time series? The method described here—using regular time intervals with average pooling (e.g., hourly aggregation)—is commonly applied to regular time series. What irregularity challenges does this approach address, and which experiments support this claim?

See also Weaknesses.

---

> ### Author Response · Authors · 2024-11-20
>
> Thank you for recognizing our work and providing insightful feedback. Your comments mainly focus on the need for additional analysis, clarifying the relationship between our work and mTAND, and revisiting Warpformer. In our responses, we have highlighted the sections where these analyses are located, revised some text to emphasize the fundamental differences between our work and mTAND, and included a review of Warpformer along with adding it as a baseline for comparison. We hope our responses address your concerns and improve your evaluation of our paper. If you have further questions, please feel free to reach out. Below are our detailed responses.
>
> ### **W1: There are some key missing analyses:**
>
> a) Comparing multi-scale modeling with single-scale modeling to quantify the benefits. An assessment of the optimal number of scales, along with the computational cost added by each scale.
>
> **A1:  We have already conducted a comparison between multi-scale and single-scale approaches in Table 6 of subsection 4.5.** The third row in Table 6 represents the single-scale results, which we found to be inferior to the performance of the full model.
>
> Searching for the optimal number of scales is time-consuming and labor-intensive. In Appendix subsection F.1, MUSICNET PARAMETERS, we explain that **our approach avoids manual selection.** **Based on the observed timestamps in each dataset**, we ensure that in the $(L-1)$-th layer, most windows contain at least one sampling point, while the $L$-th layer represents the original data layer. Under this setup, we are able to achieve good performance while saving computational resources.
>
> b) Evaluating the LSP-DTW correlation matrix against inter-variable self-attention, as seen in prior work (e.g., Warpformer), to confirm its advantage over attention-based correlation modeling.
>
> **A2: In Subsection 4.4, *Correlation Results*, we conduct a comprehensive comparison with prior work to demonstrate the necessity, effectiveness, and efficiency of the correlation matrix.**
>
> The inter-variable self-attention in Warpformer can be viewed as a learnable correlation matrix derived through attention, as shown in the 5th row of Table 4, which does not perform well in our model. While determining the best correlation calculation method for all models is beyond the scope of this work, our focus is on identifying the most suitable approach for our specific model.
>
> c) The paper mischaracterizes Warpformer as a multi-scale model for regularly sampled time series. And it should be included as an important baseline to be compared with.
>
> **A3:** We mistakenly cite a paper by an author with the same surname. We have removed this reference from the current sentence and revisited Warpformer appropriately in the *Related Work* section as follows
>
> **“As far as we know, [1] and [2] are among the earlier works on multi-level ISMTS learning. [1] addresses multi-resolution signal issues by distributing signals across specialized branches with different resolutions, where each branch employs a Flexible Irregular Time Series Network (FIT) to process high- and low-frequency data separately. [2], on the other hand, is a transformer-based model that stacks multiple Warpformer layers to produce multi-scale representations, combining them via residual connections to support downstream tasks. These works typically focus on either specific tasks or particular model architectures. In contrast, our design philosophy originates from ISMTS characteristics rather than being tied to a specific feature extraction network structure. Warpformer emphasizes designing a specific network architecture but involves high computational costs and requires manually balancing the trade-off between the number of scales and the dataset. These are challenges that our MuSiCNet avoids entirely.”**
>
> Since Warpformer uses a different benchmark than the one followed in our experiments, we **reproduced Warpformer's classification results on the benchmark dataset we used**. The results are as follows and have been included in the paper.
>
> | model | P19 |  | P12 |  | PAM |  |  |  |
> | --- | --- | --- | --- | --- | --- | --- | --- | --- |
> |  | AUROC | AUPRC | AUROC | AUPRC | Accuracy | Precision | Recall | F1 score |
> | Warpformer | 88.8±1.7  | 55.2±3.9 | 83.4±0.9 | 47.2±3.7 | 94.3±0.6 | 95.8±0.8 | 94.8±1.0 | 95.2±0.6 |
> | MuSiCNet | 86.8±1.4 | 45.4±2.7 | 86.1±0.4 | 54.1±2.2 | 96.3±0.7 | 96.9±0.6 | 96.9±0.5 | 96.8±0.5 |
>
> [1] Singh B P, et al. Multi-resolution networks for flexible irregular time series modeling (multi-fit)[J]. arXiv preprint arXiv:1905.00125, 2019.
>
> [2] Zhang J,  et al. Warpformer: A multi-scale modeling approach for irregular clinical time series[C]//Proceedings of the 29th KDD. 2023: 3273-3285.

---

> > ### Author Response · Authors · 2024-11-20
> > **Official Comment by Authors_2**
> >
> > ### **Q1:** Can MuSiCNet be viewed as a hierarchical extension of mTAND, with the addition of a correlation matrix to capture inter-variable similarity? & **Q2:** The approach to handling irregularity in MuSiCNet closely resembles that of mTAND, aside from the use of DTW. Are there any other unique design elements aimed at better adapting to irregular data?
> >
> > **A4:**  Here, we address Q1 and Q2 together.
> >
> > **We must emphasize that our work fundamentally differs from mTAND.**
> >
> > While mTAND aims to design a model specifically for handling irregular sampling by aggregating intra-series information to learn the fixed-dimensional feature.
> >
> > In contrast, we present a novel perspective: irregularity is essentially relative in some senses. **We focus on the data itself, utilizing the perspective of relative regularity to iteratively refine and complement the representations of the original irregular series, thereby addressing the irregularity issue.**
> >
> > We highlight the importance of introducing multi-level learning and capturing intra-instance relationships for ISMTS, particularly considering the sparse (see Table 7 in the appendix) yet valuable observations in such datasets.
> >
> > Under this framework, **the generated coarse-grained regular series effectively mitigate inter-series misalignments** and reduce the influence of the attention weights diminishing with distance between observations, which is a prior issue in most modeling intra-series irregularities methods. Additionally, **coarse-grained regular series mitigates modeling difficulties associated with intra-irregularities**. Our model is capable of progressively learning from relatively regular to irregular series, which aligns closely with the intuition of data-level curriculum learning [3].
> >
> > This contribution represents a key distinction from mTAND and underscores one of our significant innovations.
> >
> > ### **Q3:** What specific challenges does hierarchical modeling pose for irregularly sampled time series compared to regular time series? The method described here—using regular time intervals with average pooling—is commonly applied to regular time series. What irregularity challenges does this approach address, and which experiments support this claim?
> >
> > **A5:**
> >
> > This issue can actually be addressed by the explanation provided above in **A4**. Here, we additionally clarify the use of average pooling in ISMTS.
> >
> > Taking a sequence of length 16 as an example, its observation timestamps are "**1** 2 3 4 5 6 **7** **8** 9 10 11 12 13 14 **15** 16," and we assume the observed values are the same as the timestamps. Bold indicates the presence of observed variables. After one sampling with a window size of 4 and a sampling rate of 4, the sequence becomes "**1** **7.5** ? **15**," where "?" denotes unobserved timestamps. After the second sampling, the sequence becomes "**7.83**."
> >
> > Moreover, average pooling is merely a simple strategy we chose, and it can effectively serve the purpose we require. If simple methods can effectively address the problem, we adhere to Occam's Razor, avoiding unnecessary additional strategies. However, since standard DTW methods cannot be effectively applied to ISMTS data, we specifically designed LSP-DTW for this purpose.
> >
> > **Minor suggestion:** Address minor typos, such as the use of \citep{} and \cite{}.
> >
> > **A6:** We have made the revisions.
> >
> > [3] Wang X, Chen Y, Zhu W. A survey on curriculum learning[J]. IEEE transactions on pattern analysis and machine intelligence, 2021, 44(9): 4555-4576.

---

### Note · Authors · 2024-11-30

I have read and agree with the venue's withdrawal policy on behalf of myself and my co-authors.